# Netrin-1 inhibits the attachment and internalization of Hepatitis B virus for hepatocyte infection

Ying Wang[1ᵒ], Kazuhisa Murai[1ᵒ], Atsuya Ishida[1], Narumi Kawasaki[1], Kazuyuki Kuroki[2], Ying-Yi Li[2], Yuga Sato[1], Yutaro Miura[1], Kureha Takara[1], Lianghao Kong[1], Tetsuro Shimakami[2], Kouki Nio[2], Yuichiro Higuchi[3], Hiroshi Suemizu[3], Satoru Ito[4], Hiroshi Yanagawa[4], Shuichi Kaneko[2], Taro Yamashita[2], Masao Honda[1,2]*

1 Department of Clinical Laboratory Medicine, Kanazawa University Graduate School of Medical Sciences, Kanazawa, Japan, 2 Department of Gastroenterology, Kanazawa University Graduate School of Medical Sciences, Kanazawa, Japan, 3 Central Institute for Experimental Medicine and Life Science, Kawasaki, Japan, 4 Purotech Bio Inc., Kanagawa, Japan

ᵒ Ying Wang and Kazuhisa Murai contributed equally to this work.
* mhonda@m-kanazawa.jp

## Abstract

Netrin-1, a secreted laminin-related protein, is increasingly recognized for its role in viral pathogenesis, alongside its established functions in neural guidance and immune regulation. We previously identified endothelial lipase (LIPG) as a host factor that facilitates hepatitis B virus (HBV) attachment via heparan sulfate proteoglycans (HSPGs) and/or the sodium taurocholate cotransporting polypeptide (NTCP). Through LIPG-based screening, we identified Netrin-1 as an LIPG-interacting protein, and synthetic peptides derived from Netrin-1 sequences exhibited potent anti-HBV activity. In primary human hepatocytes, Netrin-1 demonstrated antiviral activity against HBV, and in HepG2-NTCP-YFP cells, it inhibited viral attachment and internalization. Mechanistically, Netrin-1 binds to LIPG through heparin-binding motifs in its V and C domains, disrupting LIPG-HBV interactions and displacing LIPG from HSPGs. Furthermore, Netrin-1 binds to the extracellular domain of epidermal growth factor receptor (EGFR), abrogating NTCP-EGFR complex formation and inhibiting EGFR dimerization and phosphorylation, independently of HSPGs. *In vivo*, recombinant Netrin-1 suppressed the viral infection in humanized hepatocyte chimeric mice. These findings establish Netrin-1 as a multifunctional host factor that interferes with HBV entry, supporting the development of Netrin-1-based therapeutic strategies.

## Author summary

Hepatitis B virus (HBV) is a major global cause of chronic liver disease, and current treatments rarely achieve complete viral elimination. In this study, we identified a new function for Netrin-1, a protein originally known for its role in

**Data availability statement:** All data are in the manuscript and/or supporting information files.

**Funding:** This research was supported by the Japan Agency for Medical Research and Development (Grants: JP24fk0310514 to M.H; JP25fk0310539 to M.H). The funder had no role in study design, data collection and analysis, decision to publish, or preparation of the manuscript.

**Competing interests:** The authors have declared that no competing interests exist.

guiding nerve cells, in blocking HBV infection of liver cells. Our previous work implicated endothelial lipase (LIPG) as a host factor promoting HBV entry, and we subsequently discovered that Netrin-1 interacts with LIPG and recombinant Netrin-1 suppressed the viral infection. In this study, we investigated the precise roles of Netrin-1 in HBV infection. Netrin-1 interferes with HBV entry through two independent pathways: one involving LIPG, which facilitates viral attachment, and another involving epidermal growth factor receptor, which is required for viral internalization. By blocking both steps of viral entry, Netrin-1 shows promise as a potential antiviral agent. These findings uncover a novel antiviral role for Netrin-1 and suggest new therapeutic strategies for chronic hepatitis B.

## Introduction

Hepatitis B virus (HBV) remains a major global health concern, as chronic infection can lead to severe liver disease, including fibrosis, cirrhosis, and hepatocellular carcinoma [1]. Despite the availability of effective vaccines, HBV continues to persist in millions of individuals worldwide [2]. Understanding how HBV interacts with host cellular factors during the entry process is essential for the development of novel antiviral strategies aimed at blocking infection at the earliest stage.

HBV belongs to the Hepadnaviridae family and exhibits strict hepatotropism, relying on multiple host factors to mediate its entry into hepatocytes. The process begins with the virus attaching to heparan sulfate proteoglycans (HSPGs) on the cell surface, followed by a high-affinity interaction with sodium taurocholate co-transporting polypeptide (NTCP), which serves as the primary functional receptor for HBV. Upon receptor binding, the virus is internalized via endocytosis, leading to membrane fusion and the subsequent release of its genome into the host cytoplasm. However, increasing evidence suggests that additional host factors influence HBV uptake and intracellular trafficking [3,4].

Previously, we identified endothelial lipase (LIPG) as a potential modulator of HBV infection. While LIPG is primarily recognized for its role in lipid metabolism, our findings suggest that it may influence viral entry by enhancing HBV binding to HSPGs or NTCP [5,6]. Furthermore, through *in vitro* virus screening using LIPG as bait, Netrin-1 was identified as a potential interacting protein, and synthetic peptides based on the Netrin-1 sequence inhibit HBV infection [7]. Netrin-1 is a secreted protein primarily known for its role in axonal guidance and immune regulation [8–11]. Although the potential of Netrin-1-related synthetic peptides to inhibit HBV infection has been demonstrated *in vitro* and *in vivo* [7], the functional relevance of Netrin-1 for HBV infection had not been elucidated fully. In this study, we investigated the precise roles of Netrin-1 in HBV infection, focusing on its impact on viral entry including attachment and internalization. Our data reveal a previously unrecognized relationship between Netrin-1 and LIPG in HBV infection, providing a foundation for the development of Netrin-1-oriented therapeutic approaches to prevent HBV infection.

## Results

### Netrin-1 inhibits HBV infection by suppressing HBV entry

We previously reported that LIPG upregulates HBV infection by facilitating HBV attachment to the cell membrane [6]. Then, by using an *in vitro* virus screening assay, we identified Netrin-1 as a binding partner of LIPG [7]. Netrin-1 was initially recognized as an axon guidance molecule and has been reported recently to play diverse roles in development and various pathologies [10]. To confirm the role of Netrin-1 in HBV infection, primary human hepatocytes (PXB cells) were transduced with a recombinant lentivirus encoding short hairpin RNA (shRNA) targeting Netrin-1 (shNetrin-1) or control shRNA (Fig 1A). Netrin-1 mRNA expression was significantly repressed by shNetrin-1 (Fig 1B), and HBV-DNA levels, as determined by quantitative PCR (qPCR), were substantially increased after infection for 10 days (Fig 1C). Southern blotting analysis showed a clear increase in the relaxed circular and covalently closed circular (ccc) forms of HBV DNA following Netrin-1 knockdown (Fig 1D upper). In addition, the expression of HBV core protein was substantially increased (Fig 1D lower). To verify the importance of Netrin-1 as an interacting partner of LIPG over the other binding partners of LIPG, 43 enriched candidate partners of LIPG were identified, and five binding partners were selected for functional evaluation. Suppression of these proteins by using an shRNA-lentivirus system revealed that Netrin-1 prominently influenced HBV infection in PXB cells (S1 Fig). Therefore, Netrin-1 was considered to be the most functionally important binding partner of LIPG for HBV infection.

To examine the effect of Netrin-1 on HBV replication after infection, HepG2.2.15 cells were transfected with a Netrin-1 expression vector or incubated with recombinant Netrin-1 protein (Fig 1E and 1F). After 5 days, there was no clear decrease in HBV-DNA, indicating that Netrin-1 had little effect on HBV replication (Fig 1G and 1H).

To examine the effect of Netrin-1 on the entry step of HBV in detail, we utilized newly established HepG2-NTCP-YFP cells that express the HBV receptor NTCP fused with yellow fluorescent protein (YFP) (Materials and Methods). HepG2-NTCP-YFP cells were transfected with a Netrin-1 expression vector, and after 3 days, HBV attachment (4°C for 1.5 h) and internalization (37°C for 6 h following attachment) assays were performed (Fig 1I and 1J). Netrin-1 overexpression significantly repressed HBV attachment and internalization (Fig 1K and 1L). To verify these findings further, we established doxycycline (Dox)-inducible Netrin-1-overexpressing cells (Fig 1M). Netrin-1 was expressed in a Dox concentration-dependent manner (Fig 1M), and no cellular damage was observed at the maximum concentration of 25 ng/mL (Fig 1N). Interestingly, HBV attachment and internalization were significantly repressed in a Netrin-1 dose-dependent manner (Fig 1O and 1P).

Netrin-1 is a secreted protein and interacts with HSPGs on the cell surface. To examine whether Netrin-1 was secreted from Dox-inducible Netrin-1-overexpressing cells and inhibited HBV attachment and internalization, we utilized a Transwell co-culture system. Dox-inducible Netrin-1-overexpressing HepG2-NTCP-YFP cells were seeded in the upper chamber of a 0.4-μm pore Transwell insert, and wild-type HepG2-NTCP-YFP cells were seeded in the lower chamber (Fig 1Q), allowing secreted proteins to diffuse between both compartments. After Dox treatment for 72 h, a Dox dose-dependent increase of Netrin-1 in the culture medium was observed, while there was no significant change in the concentration of LIPG (S2 Fig). Importantly, a Dox dose-dependent increase of Netrin-1 was detected in the membrane fraction and whole cell lysate of cells in the lower chamber (Fig 1R). After removing the upper chamber, cells from the lower chamber were subjected to HBV inoculation. As a result, HBV attachment (Fig 1S) and internalization (Fig 1T) were significantly reduced in these cells compared to control co-cultures without the Dox-dependent induction of Netrin-1. These findings indicate that Netrin-1 is secreted from hepatocytes and inhibits HBV entry in a *trans*-acting manner. Moreover, recombinant Netrin-1, freshly added to the culture medium, also inhibited HBV attachment and internalization in an approximately dose-dependent fashion (S3 Fig).

We evaluated the expression of LIPG and Netrin-1 over the course of HBV infection in PXB cells and HepG2-derived HBV permissive cells. As Netrin-1 is expressed at a low level in HepG2-NTCP-YFP cells, we utilized HepG2-NTCPsec+ cells, which express Netrin-1 at an easily detectable level [12]. In the early phase of HBV infection (from day 3 to day 6),

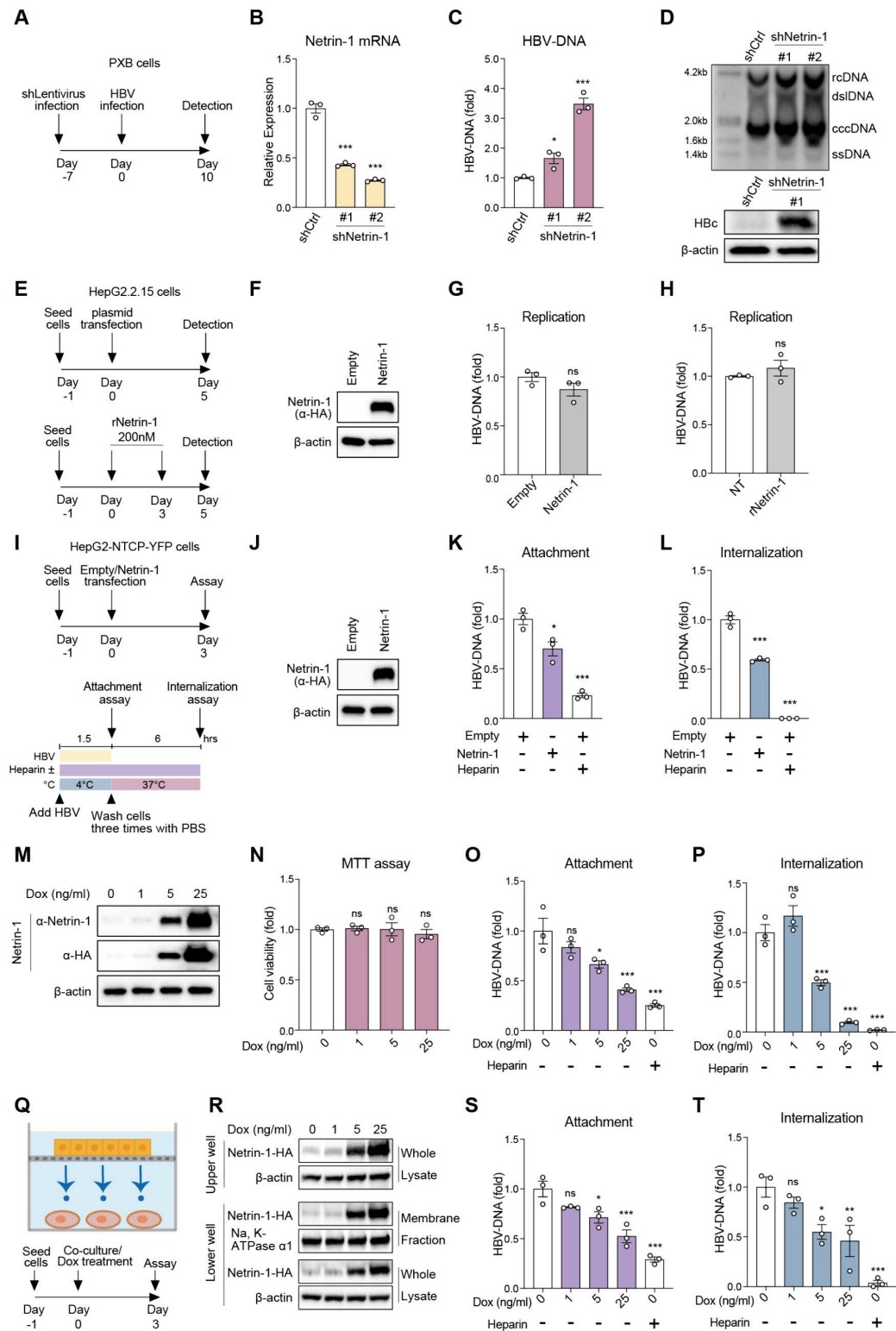

**Fig 1. Netrin-1 inhibits HBV infection by suppressing HBV entry.** A: Schematic schedule of shRNA-lentivirus transduction and HBV infection in PXB cells. B: Quantitative real-time PCR analysis of Netrin-1 mRNA in PXB cells transduced by shNetrin-1-lentivirus at 10 days post-HBV infection. C: qPCR analysis of intracellular HBV DNA at 10 days post-HBV infection. D: Southern blotting analysis of intracellular HBV DNA in PXB cells transduced by shNetrin-1-lentivirus at 10 days post-HBV infection. The positions of relaxed circular DNA (rcDNA), double-stranded liner (dslDNA), cccDNA, and

single-stranded DNA (ssDNA) are indicated (upper). Western blotting analysis of intracellular hepatitis B core protein (HBc) and β-actin (loading control) (lower). E: Experimental design for the HBV replication assay using plasmid transfection (upper) or recombinant Netrin-1 (lower). F: Western blotting of Netrin-1-HA and β-actin in HepG2.2.15 cells transfected with empty plasmid or Netrin-1-HA plasmid. Netrin-1 was detected using an α-HA antibody. G and H: To evaluate HBV replication, extracellular HBV DNA from HepG2.2.15 cells was quantified at 5 days post-transfection (G) or after 5 days of incubation with 200 nM recombinant Netrin-1 (rNetrin-1) (H). I: HBV attachment and internalization assays using HepG2-NTCP-YFP cells transfected with empty plasmid or Netrin-1-HA plasmid. J: Western blotting of Netrin-1-HA and β-actin post-transfection, as described in panel I. Netrin-1 was detected using an α-HA antibody. K: HBV attachment assay: the cells were exposed to HBV at 4°C for 1.5 h; surface-bound HBV DNA was quantified by qPCR. L: HBV internalization assay: HBV-attached cells incubated at 37°C for 6 h, treated with trypsin, and intracellular HBV DNA was measured. M–P: Dox-inducible Netrin-1-overexpressing HepG2-NTCP-YFP cells were incubated with 0, 1, 5, or 25 ng/mL Dox for 72 h to induce Netrin-1 expression, followed by HBV entry assays. Western blotting of Netrin-1 (M), MTT assay for cell viability (N), HBV attachment (O), and internalization (P). Q: Illustration and schematic timeline of the Transwell co-culture system. Co-culture was initiated at 1 day after cell seeding. Dox-inducible Netrin-1-overexpressing HepG2-NTCP-YFP cells were seeded in the upper well, while HepG2-NTCP-YFP cells were seeded in the lower well. Cells were treated with 0, 1, 5, or 25 ng/mL Dox for 72 h to induce Netrin-1 expression, followed by western blotting analysis (R) of the upper and lower cells and HBV entry assays (S: attachment; T: internalization). R: Western blotting analysis. For the upper well cells, whole cell lysates were analyzed for Netrin-1-HA and β-actin (loading control). For the lower well cells, whole cell lysates and membrane fractions were analyzed. Whole cell lysates were probed for Netrin-1-HA and β-actin; membrane fractions were analyzed for Netrin-1-HA and Na$^+$/K$^+$-ATPase α1 (loading control). α-HA antibody (for Netrin-1-HA), β-actin, and Na$^+$/K$^+$-ATPase α1 antibodies were used. S and T: For HBV attachment (S) and internalization (T) assays, the upper wells were removed, and the cells in the lower well were subjected to the assays. Data are the mean ± standard error of the mean ($n = 3$). Statistical analysis was performed using a two-tailed unpaired $t$-test (G, H) or two-way ANOVA with Tukey's test (B, C, K, L, N, O, P, S, T). ****$p < 0.0001$, ***$p < 0.001$, *$p < 0.05$, ns = not significant.

the expression of Netrin-1 and LIPG did not change according to HBV infection in PXB cells, although Netrin-1 expression decreased over the course of the culture period regardless of HBV infection (S4A Fig). The concentrations of LIPG and Netrin-1 as determined by enzyme-linked immunosorbent assays (ELISAs) in culture medium correlated well with those detected by western blot analysis (S4A–S4C Fig). In the delayed phase of infection (from day 8 to day 12), LIPG expression was induced more in HBV-infected cells, as previously reported [6]. However, expression of Netrin-1 was reduced by HBV infection in PXB cells (S4D Fig). In HepG2-NTCPsec+ cells, expression of Netrin-1 was also reduced by HBV infection, while LIPG levels remained unchanged in the cells and culture medium (S4E–S4F Fig). Thus, expression of Netrin-1 could be reduced by HBV infection, although its general expression was gradually reduced over the culture period.

## Netrin-1 interacts with LIPG and suppresses HSPG-dependent HBV attachment

In the process of entry into hepatocytes, HBV attaches to HSPGs on their surface with low affinity and then interacts with NTCP in a high-affinity manner [4]. In addition, HBV can attach to hepatocytes in an HSPG-independent manner as a preS1-probe was shown to interact directly with NTCP without any interference by heparin (Fig 2A) [6]. LIPG might stimulate HSPG-dependent and HSPG-independent HBV attachment, as reported previously [6].

To explore the role of Netrin-1 in HSPG-independent HBV attachment, we examined its effect on the attachment of the preS1-probe to the surface of HepG2-NTCP-YFP cells (Fig 2B). Interestingly, there was no decrease in preS1-probe attachment to the surface of Dox-induced Netrin-1-overexpressing cells (Fig 2B). These results suggest that Netrin-1 has no effect on HSPG-independent HBV attachment. To examine these findings further, we performed an HBV attachment assay after heparanase treatment of the cells. Heparanase, a mammalian endo-β-D-glucuronidase, cleaves the glycosaminoglycan heparan sulfate side chains of HSPG [6,13] to eliminate HSPG-dependent HBV attachment. Heparanase treatment decreased total HBV attachment to approximately 25%; however, there was no further decrease in HBV attachment despite an increase of Netrin-1 expression (Fig 2C). Therefore, Netrin-1 inhibits HSPG-dependent HBV attachment and might be irrelevant for HSPG-independent HBV attachment.

Netrin-1 is a member of a family of laminin-related secreted proteins and is thought to be involved in axon guidance and cell migration during development [8]. Netrin-1 consists of a highly conserved N-terminal laminin domain (domain VI), domain V with three laminin-type epidermal growth factor (EGF)-like repeats (LE1, LE2, and LE3), and a positively charged C-terminal Netrin-like domain (Fig 2D). Netrin-1 functions as a pleiotropic ligand by interacting with different

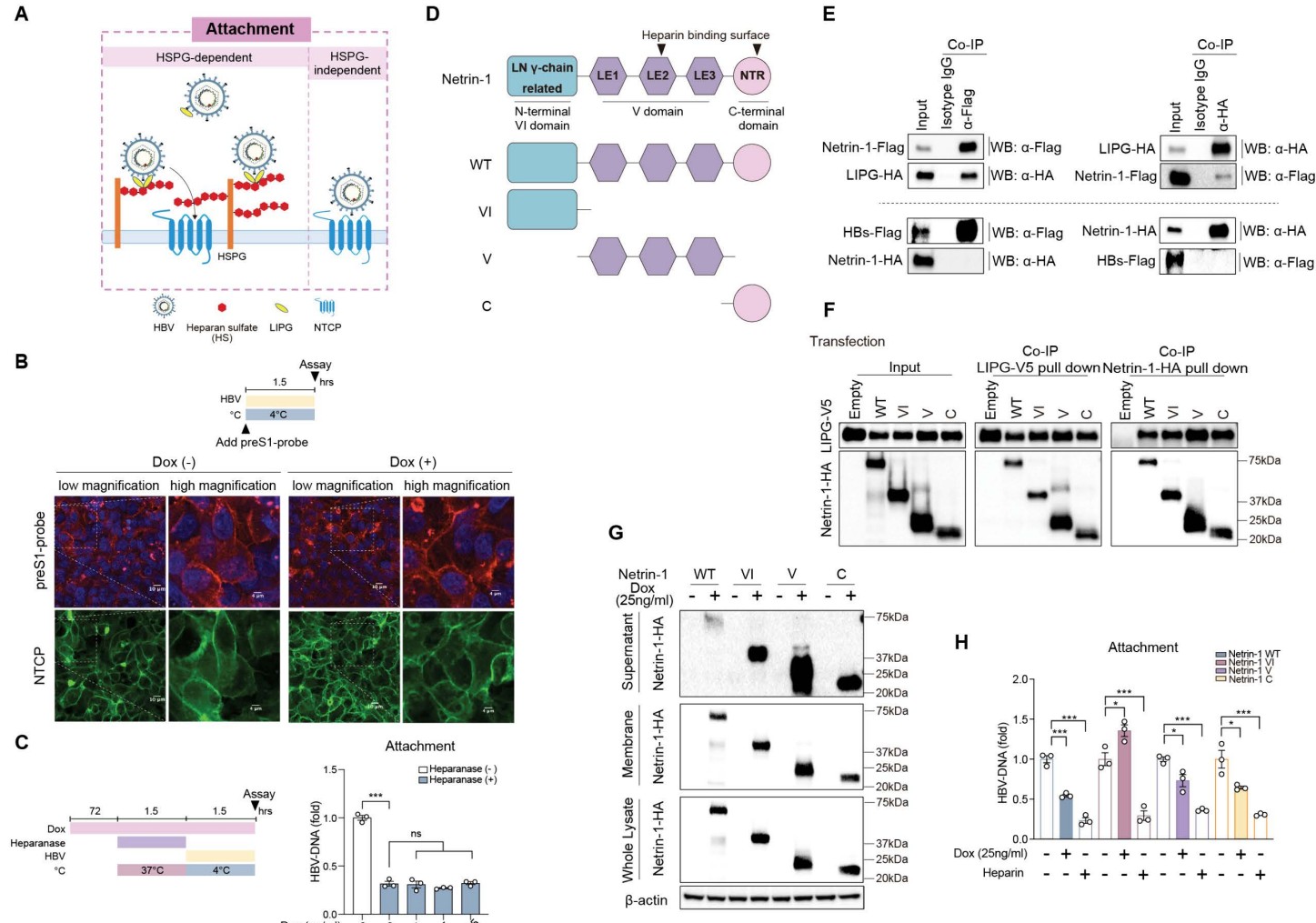

**Fig 2. Netrin-1 interacts with LIPG and suppresses HSPG-dependent HBV attachment.** A: Schematic of HBV attachment to hepatocytes. B: Dox-inducible Netrin-1-overexpressing HepG2-NTCP-YFP cells were incubated with or without 25 ng/mL Dox for 72 h and then inoculated with the preS1-probe at 4°C for 1.5 h to allow surface attachment (upper). After fixation, fluorescence was observed by confocal microscopy. Representative images (preS1-probe, red; NTCP-YFP, green; nucleus, blue), shown at low (scale bar, 10 μm) and high (scale bar, 4 μm) magnification (lower). C: Dox-inducible Netrin-1-overexpressing HepG2-NTCP-YFP cells were incubated with 0, 1, 5, or 25 ng/mL Dox for 72 h to induce Netrin-1 expression. The cells were then pretreated with heparanase (0.5 μg/mL) at 37°C for 1.5 h to remove HSPG heparan sulfate side chains, followed by incubation with HBV at 4°C for 1.5 h. Surface-bound HBV DNA was assessed by qPCR. D: Structure of wild-type and truncated Netrin-1 variants. LE1, LE2, and LE3, laminin-type epidermal growth factor-like repeats; LN, laminin domain; NTR, netrin-like domain. E: HepG2 cells were transfected with Netrin-1-Flag and LIPG-HA (upper) or HBs-Flag and Netrin-1-HA (lower) plasmids. At 48 h after transfection, the cells were immunoprecipitated with α-Flag antibody (for Netrin-1) or α-HA antibody (for LIPG) and detected using α-HA antibody (for LIPG) or α-Flag antibody (for Netrin-1) (upper), immunoprecipitated with α-Flag antibody (for HBs) or α-HA antibody (for Netrin-1) and detected by α-HA antibody (for Netrin-1) or α-Flag antibody (for HBs) (lower). F: HepG2 cells were co-transfected with LIPG-V5 and wild-type or truncated Netrin-1-HA plasmids. At 48 h after transfection, the cells were immunoprecipitated with α-V5 antibody (for LIPG) or α-HA antibody (for Netrin-1) and detected by α-V5 antibody (for LIPG) or α-HA antibody (for Netrin-1). G and H: Dox-inducible HepG2-NTCP-YFP cells expressing wild-type or truncated Netrin-1 were cultured with or without 25 ng/mL Dox for 72 h, followed by detection. Culture supernatants, membrane fractions, and whole cell lysates were subjected to western blotting analysis (G), and an HBV attachment assay was performed (H). Data are the mean ± standard error of the mean ($n = 3$). Statistical analysis was performed using two-way ANOVA with Tukey's test. ****$p < 0.0001$, ***$p < 0.001$, *$p < 0.05$, ns = not significant.

receptors, with typical receptors such as UNC5B and neogenin also being expressed by hepatocytes. To examine whether Netrin-1 signaling had any effect on HBV entry, we introduced mutations at the critical residues of Netrin-1 for binding to its receptors [14]. L111A is a critical mutation for the binding of Netrin-1 to the DCC and neogenin receptors, and R348-R349-R351A are critical mutations for the binding of Netrin-1 to the UNC5B receptor (S5 Fig). Overexpression of these receptor-binding mutants inhibited HBV attachment and internalization, similar to wild-type Netrin-1, suggesting that Netrin-1 signaling in cells is irrelevant for HBV entry (S5 Fig).

Netrin-1 is a secreted heparin/heparan sulfate-binding protein and interacts with HSPGs through heparin/heparan sulfate-binding sites possibly located in the V [15] and C [16] domains (Fig 2D). Although we previously reported that Netrin-1 is a binding partner of LIPG [7], the detailed interactions of Netrin-1 and LIPG have not been clarified. To explore the functional role of Netrin-1 in HSPG-dependent HBV attachment, we examined the interactions of Netrin-1 and LIPG in more detail. Super-resolution confocal microscopy showed that Netrin-1 was expressed on the cell surface and in the cytoplasm, whereas LIPG was mainly expressed on the surface of HepG2 cells. Clear co-localization of Netrin-1 and LIPG was observed on the cell surface (S6 Fig). Co-immunoprecipitation (Co-IP) analysis of Netrin-1 and LIPG showed the binding of these two proteins (Fig 2E). We previously showed that LIPG binds to HBV large surface proteins (LHBs) [6], but here, we found that Netrin-1 did not bind to LHBs (Fig 2E). We searched for the domains of Netrin-1 responsible for its binding to LIPG. Co-IP analysis showed that each domain of Netrin-1 (VI, V, and C) independently bound to LIPG in HepG2 cells (Fig 2F), similar to its binding to the UNC5 receptor [17]. These findings were also confirmed using an *in vitro* cell-free translation system, indicating that each domain of Netrin-1 (VI, V, and C) bound directly to LIPG and this binding was not dependent on the presence of the cell membrane (S7 Fig). To explore which domain of Netrin-1 functionally influenced HBV attachment, we established Dox-inducible HepG2-NTCP-YFP cells that overexpressed individual Netrin-1 domains (VI, V, or C). Western blotting analysis confirmed that all domains were detected and comparably expressed in both the whole cell lysate and membrane fraction, with all three domains showing extremely high stability compared to wild-type Netrin-1 in culture supernatants (Fig 2G). Interestingly, domains V and C, but not domain VI, suppressed HBV attachment to a similar extent as wild-type Netrin-1 did (Fig 2H). These findings suggest that the V and C domains harbor specific functional properties required for the inhibition of HBV attachment, which are not shared by the VI domain.

## V and C domains of Netrin-1 interact with the heparin-binding site (HBS) of LIPG

Since the V and C domains of Netrin-1 independently interacted with LIPG and retained the ability to reduce HBV attachment, we hypothesized that their interactions with LIPG play a role in the suppression of HBV attachment. Netrin-1 has been characterized as possessing heparin/heparan sulfate-binding sites in the V (LE2 subdomain) and C domains, both consisting of positively charged amino acid patches [15,16] (Fig 3A and 3B). On the other hand, LIPG exerts its lipase activity by anchoring to HSPGs via its heparin-binding motif (amino acids 325–337), followed by bridging and facilitating the uptake of high-density lipoprotein into cells [18]. Full-length LIPG (68 kDa) can be inactivated by cleavage at the arginine 330 site within the heparin-binding motif by a proprotein convertase, resulting in an N-terminal 40-kDa fragment (N-terminal domain; NTD) and a C-terminal 28-kDa fragment (C-terminal domain; CTD) (Fig 3C) [19]. To explore the regions of LIPG responsible for binding to the V and C domains of Netrin-1, we constructed the NTD (LIPG-NTD) and CTD (LIPG-CTD) of LIPG with or without the HBS, which consists of a short stretch (10–20) of amino acids (Fig 3C). Co-IP analysis showed that the V domain interacted with LIPG-NTD, but not with LIPG-CTD (Fig 3D). The V domain did not interact with LIPG-NTD (amino acids 1–310) that did not include the HBS, indicating that the HBS of LIPG-NTD is an essential region for the interaction of the V domain with LIPG-NTD (Fig 3D). Mutation of the HBS in domain V (V mut) abolished the interaction with LIPG-NTD (Fig 3E), suggesting that the HBSs in LIPG-NTD and domain V were essential for these two interactions. On the contrary, the C domain of Netrin-1 interacted with LIPG-NTD and LIPG-CTD, irrespective of the HBS of LIPG (Fig 3F). However, a domain C mutant (C mut) with a deletion of the HBS (amino acids 580–604) (Fig 3A) showed low affinity for LIPG-CTD (Fig 3G), suggesting the importance of the interaction of the HBSs in domain

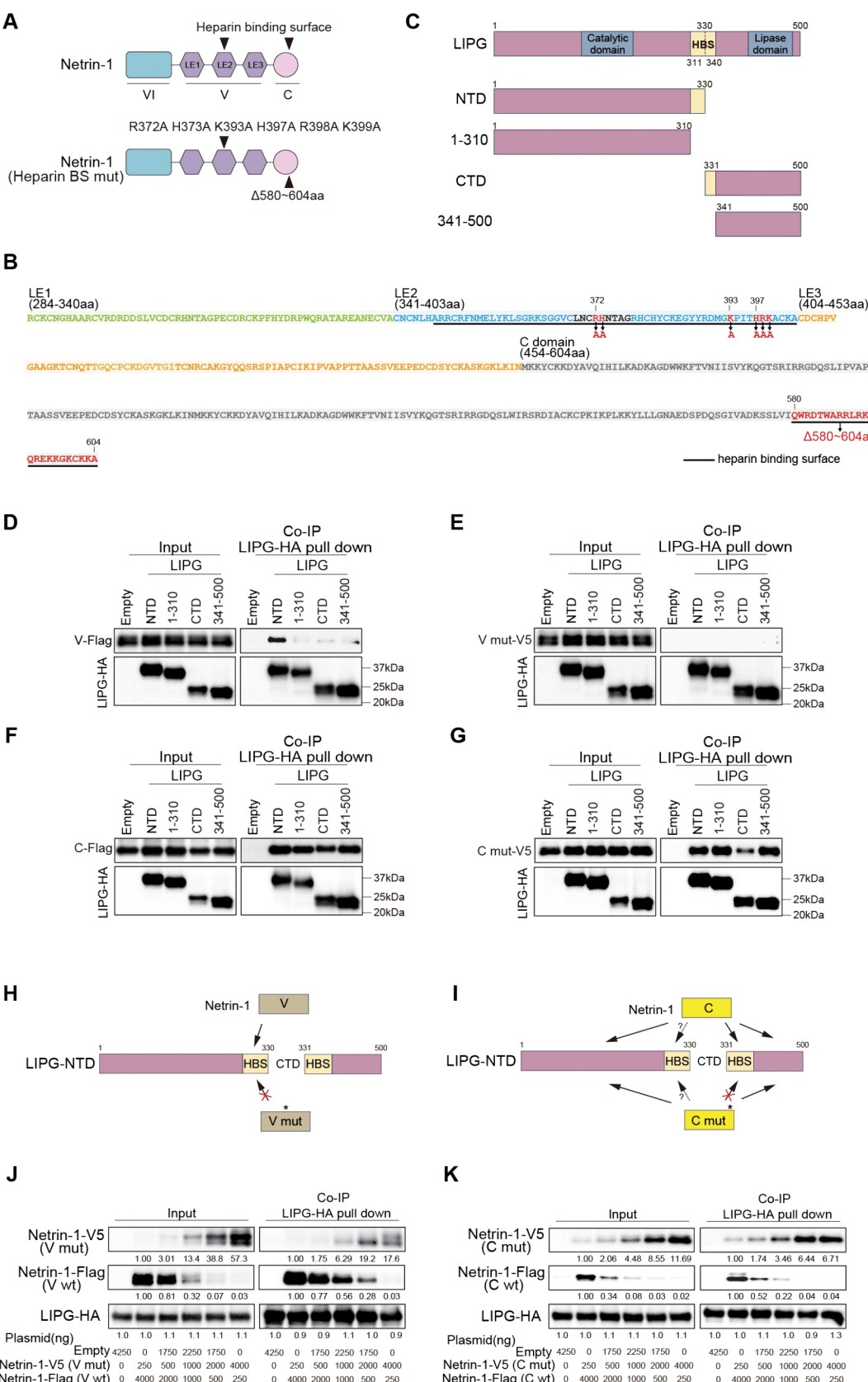

**Fig 3. V and C domains of Netrin-1 interact with the HBS of LIPG.** A and B: Schematic representation of the heparin/heparan sulfate-binding sites of Netrin-1 and the substituted amino acid mutations. C: Diagram of wild-type LIPG and cleaved NTD and CTD, with or without deletion of the heparin-binding motif. D and E: HepG2 cells were co-transfected with a series of truncated forms of LIPG-HA and wild-type Netrin-1 V domain (V-Flag) (D) or heparin/heparan sulfate-binding site mutant V domain (V mut-V5) (E) plasmids. The cells were immunoprecipitated with α-HA antibody (for LIPG) and then detected with α-Flag antibody (for V) or α-V5 antibody (for V mut). F and G: HepG2 cells were co-transfected with a series of truncated forms of LIPG-HA and wild-type Netrin-1 C domain (C-Flag) (F) or heparin/heparan sulfate-binding site mutant C domain (C mut-V5) (G) plasmids. Cells were immunoprecipitated with α-HA antibody (for LIPG) and then detected with α-Flag antibody (for C) or α-V5 antibody (for C mut). H: A model of the Netrin-1 V domain interacting with LIPG. The Netrin-1 V domain interacts with the NTD heparin-binding motif of LIPG, while the heparin/heparan sulfate-binding site mutant V domain loses this ability. I: A model of the Netrin-1 C domain interacting with LIPG. The Netrin-1 C domain interacts with all domains of LIPG, while the heparin/heparan sulfate-binding site mutant C domain loses its affinity for the CTD heparin-binding motif of LIPG, but retains its affinity for the other domains. J: Dox-inducible full-length LIPG-HA-expressing HepG2-NTCP-YFP cells were co-transfected with full-length wild-type Netrin-1-Flag, mutant Netrin-1-V5 with heparin/heparan sulfate-binding site mutation in the V domain, and pcDNA3.1(+) empty plasmids (for a total of 4,250 ng plasmid per sample). The cells were incubated with 25 ng/mL Dox to induce LIPG expression. At 48 h after transfection, the cells were immunoprecipitated with α-HA antibody (for LIPG) and then detected with α-HA antibody (for LIPG), α-Flag antibody (for wild-type Netrin-1), or α-V5 antibody (for Netrin-1 with mutation in the V domain). K: Dox-inducible full-length LIPG-HA-expressing HepG2-NTCP-YFP cells were co-transfected with full-length wild-type Netrin-1-Flag, mutant Netrin-1-V5 with heparin/heparan sulfate-binding site mutation in the C domain, and pcDNA3.1(+) empty plasmids (for a total of 4,250 ng plasmid per sample). The cells were incubated with 25 ng/mL Dox to induce LIPG expression. At 48 h after transfection, the cells were immunoprecipitated with α-HA antibody (for LIPG) and then detected with α-HA antibody (for LIPG), α-Flag antibody (for wild-type Netrin-1), or α-V5 antibody (for Netrin-1 with mutation in the C domain).

C and LIPG-CTD. These results indicate that domain V interacts with LIPG-NTD through HBSs (Fig 3H). On the other hand, the C domain interacts with LIPG-CTD through HBSs, although it can also interact with other regions of LIPG in an HBS-independent manner (Fig 3I).

To examine whether mutation of the V and C domains in Netrin-1 could change the affinity of full-length Netrin-1 to full-length LIPG, we performed a competitive Co-IP assay. Full-length Netrin-1 containing V mut had less affinity to LIPG compared to wild-type Netrin-1 (Fig 3J), while full-length Netrin-1 containing C mut had comparable affinity to LIPG (Fig 3K). Thus, different binding sites and the affinity of the V and C domains of Netrin-1 to LIPG might reflect different modes of action for the suppression of HBV attachment.

## C domain of Netrin-1 suppresses the interaction of LIPG and LHBs

We previously showed that LIPG interacts with LHBs[6]. In the present study, we determined the regions of LIPG responsible for its interactions with LHBs (Fig 4A). Co-IP analysis showed that LHBs bound to LIPG-CTD, but not to LIPG-NTD (Fig 4B). The interaction of LHBs and LIPG-CTD was almost completely lost when the HBS was deleted from LIPG-CTD (the 341–500 construct, Fig 4C). To demonstrate the direct interaction of LHBs and LIPG-CTD HBS, a biotinylated LIPG-CTD HBS peptide (amino acids 331–340; NSKMYLKTRA) was synthesized and the interaction of LHBs and LIPG-CTD HBS was analyzed using a biotinylated protein pull-down assay (Fig 4D). LHBs were successfully pulled down by the biotinylated LIPG-CTD HBS peptide, showing the direct interaction of LHBs and LIPG-CTD HBS (Fig 4D, 0 ng/μL recombinant Netrin-1). Interestingly, as the concentration of input recombinant Netrin-1 was increased, the amount of pulled-down LHBs decreased (Fig 4D). These results indicate that Netrin-1 can compete with the direct interaction of LHBs and LIPG-CTD HBS.

We next examined the effect of Netrin-1 on the interaction between wild-type LIPG and LHBs in cell-based experiments. In the condition in which LIPG interacted with LHBs (Fig 4E upper), overexpressed Netrin-1 interacted with LIPG and the interaction of LIPG and LHBs was lost (Fig 4E lower). Thus, the binding of Netrin-1 to LIPG could compete out LHBs from LIPG in cell-based conditions. To explore these findings in more detail, Netrin-1 was overexpressed in Dox-induced cell lines and LHBs were pulled down by LIPG. When Netrin-1 expression was increased, the amount of LHBs pulled down by LIPG gradually decreased (Fig 4F). Similar experiments were performed using Dox-induced IV, V, C, VI + V, and C mut-expressing cells and LHBs pulled down by LIPG were monitored. The ratio of pulled-down LHBs by LIPG compared to input LHBs (pulled LHBs/ input LHBs) is an appropriate indicator of the binding of LHBs and LIPG. The data showed that overexpression of the C domain reduced the binding of LHBs and LIPG, similar to wild-type Netrin-1,

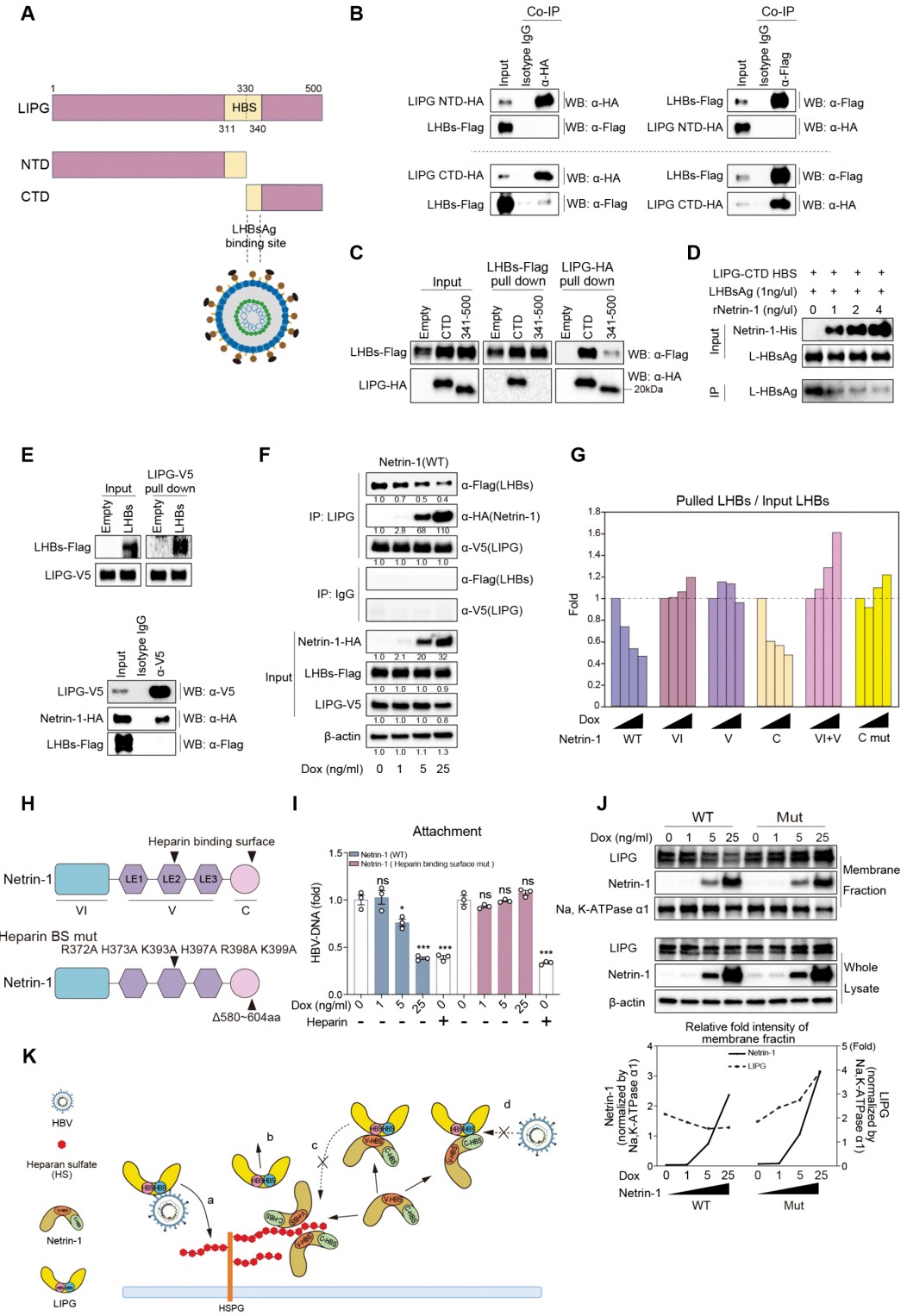

**Fig 4. C domain of Netrin-1 suppresses the interaction of LIPG and LHBs.** A: Proposed mechanism for the binding of LHBs to the C-terminal heparin-binding motif of LIPG CTD. B: HepG2 cells were transfected with LHBs-Flag and LIPG NTD-HA (upper) or LHBs-Flag and LIPG CTD-HA (lower) plasmids. At 48 h after transfection, the cells were immunoprecipitated with α-HA antibody (for LIPG NTD/CTD) or α-Flag antibody (for LHBs) and detected using an α-HA antibody (for LIPG NTD/CTD) or α-Flag antibody (for LHBs). C: HepG2 cells were co-transfected with LHBs-Flag and truncated LIPG CTD-HA plasmids, either containing or lacking the heparin-binding motif. At 48 h after transfection, the cells were immunoprecipitated with

α-Flag antibody (for LHBs) or α-HA antibody (for LIPG CTD) and then detected with α-Flag antibody (for LHBs) or α-HA antibody (for LIPG CTD). D: The C-terminal HBS of LIPG (amino acids 331–340; NSKMYLKTRA) was biotinylated at the N-terminus. For each assay, 150 μL of 0.5 mg/mL biotinylated peptide was incubated with a streptavidin-immobilized gel. Subsequently, 100 μL of 1 ng/μL recombinant LHBs, in the presence of 0, 1, 2, or 4 ng/μL recombinant Netrin-1 (rNetrin-1)-His, was added to the peptide-immobilized gel. After incubation, proteins bound to the biotinylated peptide were eluted and subjected to western blotting analysis. rNetrin-1 was detected using α-His antibody, and large HBsAg was detected using α-HBsAg antibody. E: HepG2 cells were co-transfected with LHBs-Flag and wild-type LIPG-V5 plasmids, with or without the addition of wild-type Netrin-1-HA plasmids (lower and upper, respectively). At 48 h after transfection, the cells were immunoprecipitated with α-V5 antibody (for LIPG) and then detected with α-Flag antibody (for LHBs), α-V5 antibody (for LIPG), or α-HA antibody (for Netrin-1). F and G: Dox-inducible HepG2-NTCP-YFP cells expressing wild-type or truncated forms of Netrin-1-HA were co-transfected with LHBs-Flag and wild-type LIPG-V5 expression plasmids, and then incubated with 0, 1, 5, or 25 ng/mL Dox for 48 h. The cells were harvested for immunoprecipitation with α-V5 (for LIPG) or IgG-Ctrl antibody and then detected with α-Flag antibody (for LHBs), α-HA antibody (for Netrin-1), or α-V5 antibody (for LIPG). H: Illustration of wild-type Netrin-1 and Netrin-1 with mutations in the heparin/heparan sulfate-binding sites. I: Dox-inducible HepG2-NTCP-YFP cells expressing wild-type Netrin-1 or heparin-binding-deficient mutant were incubated with 0, 1, 5, or 25 ng/mL Dox for 48 h, and then subjected to the HBV attachment assay. J: Whole cell lysates or isolated membrane fractions from samples under the same conditions as in panel H were subjected to western blotting analysis (upper). The membrane protein Na$^+$/K$^+$-ATPase α1 (ATP1A1) was used as a loading control for the membrane fraction. Band intensities of Netrin-1 and LIPG in the membrane fraction were normalized to ATP1A1 and plotted in a graph (lower). K: Depiction of Netrin-1 interfering with the LIPG-dependent enhancement of HBV attachment. (a) HBV binds to the C-terminal heparin-binding motif of LIPG through LHBs, and is then transported to HSPGs via the LIPG-HSPG interaction. Netrin-1 interferes with this process in three ways: (b) it binds to the heparan sulfate side chains of HSPGs, causing the release of LIPG from HSPGs; (c) Netrin-1 interacts with free LIPG, occupying its heparin-binding motif and preventing LIPG from anchoring to HSPGs; and (d) Netrin-1 competes with LHBs for binding to the C-terminal heparin-binding motif of LIPG. Data are the mean ± standard error of the mean ($n = 3$). Statistical analysis was performed using two-way ANOVA with Tukey's test. ****$p < 0.0001$, ***$p < 0.001$, *$p < 0.05$, ns = not significant.

while overexpression of the VI and V domains had no effect. Interestingly, Netrin-1 C mut (C domain Δ580–604 aa) did not inhibit the binding of LHBs and LIPG (Fig 4G). These data were compatible with the finding that the C domain could bind to the HBS of LIPG-CTD (Fig 3I), which also interacts with LHBs (Fig 4A).

The results so far indicate that the V domain of Netrin-1 binds to the HBS of LIPG-NTD and can interfere with the binding of LIPG to HSPGs on the cell membrane. On the other hand, the C domain of Netrin-1 binds to the HBS in LIPG-CTD and can interfere with the binding of LIPG to LHBs. Therefore, we examined the effects of HBS mutations in the Netrin-1 V and C domains on HBV attachment. An HBS mutant of Netrin-1, including the V and C domains (Fig 4H), completely lost the ability to inhibit HBV attachment (Fig 4I). Fine analysis of LIPG and Netrin-1 expression in the membrane fraction revealed that with the increase in the expression of wild-type Netrin-1, LIPG expression gradually decreased, while with the increase in the expression of mutant Netrin-1, LIPG expression gradually increased (Fig 4J). These results indicate that Netrin-1 and LIPG compete for binding to HSPGs on the cell membrane.

There are three possible mechanisms by which Netrin-1 inhibits HBV attachment. Firstly, Netrin-1 may compete out LIPG from HSPGs on the cell membrane by its heparin-binding activity (step 1, Fig 4K-b). Secondly, Netrin-1 could interact with LIPG through the HBS in the V domain and inhibit the binding of LIPG to HSPGs (step 2, Fig 4K-c). Thirdly, Netrin-1 might interact with LIPG through the HBS in the C domain and inhibit the binding of LIPG to LHBs (step 3, Fig 4K-d). Interestingly, we previously showed that LIPG-binding peptides (LIPH4-NTNBS and LIPH4-23S) [7] inhibit HBV infection and these peptides share the consensus amino acid sequence of the HBS in the V domain of Netrin-1 (S8 Fig).

## Netrin-1 interacts with epidermal growth factor receptor (EGFR) and suppresses EGFR-mediated HBV internalization

HBV, once specifically transferred to NTCP, is then internalized into the cell with EGFR, which has been reported to act as a host-entry cofactor [20]. Netrin-1 may suppress HBV internalization as well as HBV attachment (Fig 1P). To confirm the suppressive effect of Netrin-1 on HBV internalization during HBV particle infection, Dox-inducible Netrin-1-overexpressing HepG2-NTCP-YFP cells were treated with heparanase to remove the HSPG-dependent inhibitory effect of Netrin-1 on HBV attachment. Heparanase treatment decreased HBV internalization to approximately 30% compared to baseline (no Dox, with or without heparanase), and importantly, Dox-induced Netrin-1 overexpression further suppressed HBV internalization at 5 and 25 ng/mL Dox (Fig 5A).

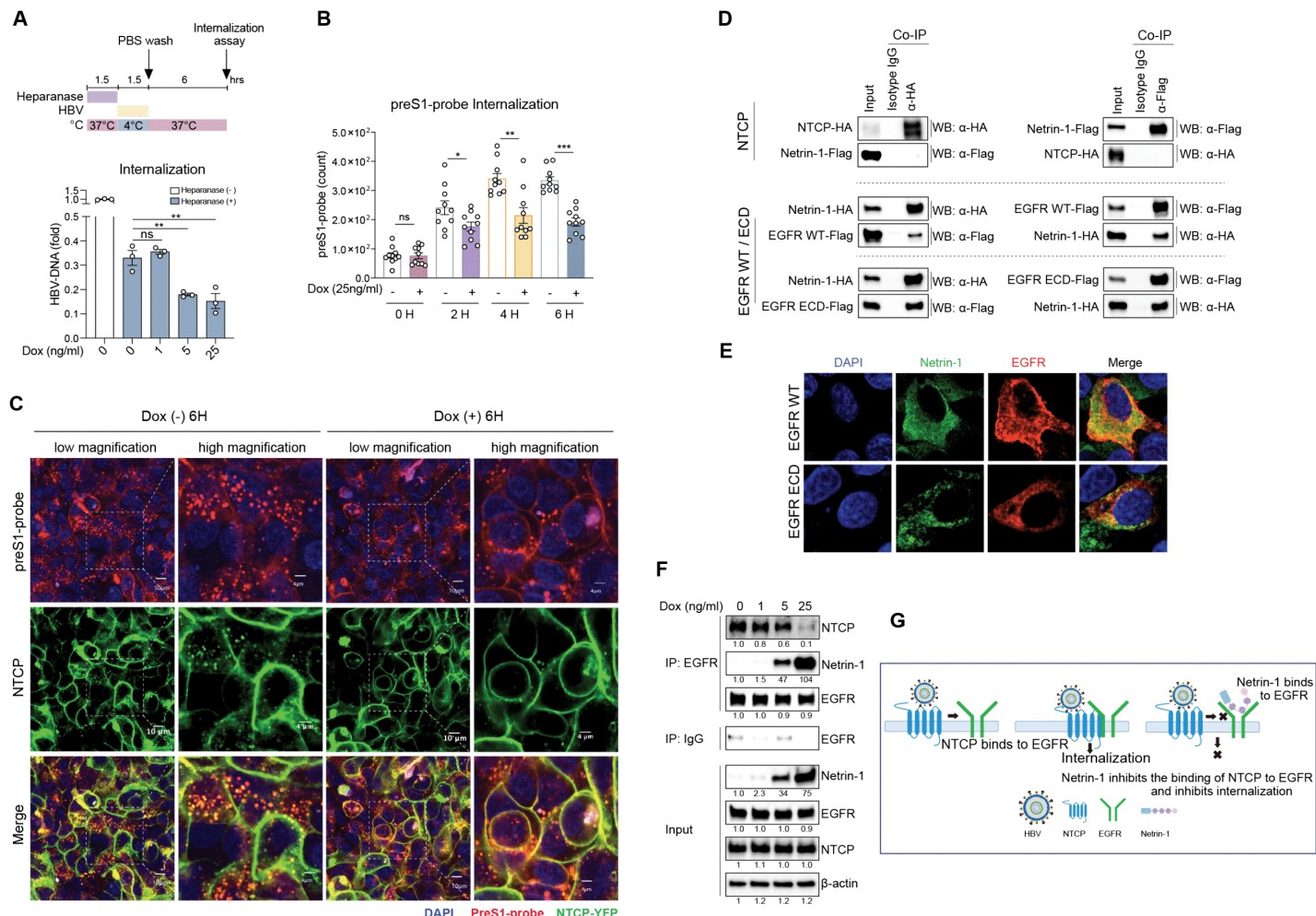

**Fig 5. Netrin-1 interacts with EGFR and suppresses EGFR-mediated HBV internalization.** A: Dox-inducible HepG2-NTCP-YFP cells expressing wild-type Netrin-1 were incubated with 0, 1, 5, or 25 ng/mL Dox for 72 h to induce Netrin-1 expression. Then, the cells were pre-treated with heparanase (0.5 µg/mL) at 37°C for 1.5 h, followed by incubation with HBV at 4°C for 1.5 h to allow HBV attachment. After washing with PBS, the cells were incubated at 37°C for 6 h to facilitate viral entry. Following 2.5% trypsin treatment to remove surface-bound virus, intracellular HBV DNA was analyzed by qPCR. B and C: Dox-inducible Netrin-1-overexpressing HepG2-NTCP-YFP cells were incubated with or without 25 ng/mL Dox for 72 h, and then inoculated with the preS1-probe at 4°C for 1.5 h to allow probe attachment to the cell surface. The preS1-probe-bound cells were incubated at 37°C to facilitate preS1-probe endocytosis, and the endocytosed fluorescence signal was detected at 0, 2, 4, and 6 h. The cells were fixed and stained with DAPI (blue) for nuclear staining. The samples were observed by confocal microscopy. The internalized preS1-probe signal was quantified using ImageJ from 10 fields of view per sample and plotted in the graph (B). (C) Representative images of the samples at 6 h (preS1-probe, red; NTCP-YFP, green; nucleus, blue; merged preS1-probe and NTCP-YFP, yellow), shown at low (scale bar, 10 µm) and high (scale bar, 4 µm) magnification. D: HepG2 cells were co-transfected with Netrin-1-Flag and NTCP-HA (upper), or Netrin-1-HA together with wild-type EGFR-Flag or EGFR-ECD-Flag (lower) overexpression plasmids. At 48 h after transfection, the cells were immunoprecipitated with α-HA antibody (for NTCP) or α-Flag antibody (for Netrin-1) and detected using an α-HA antibody (for NTCP) or α-Flag antibody (for Netrin-1) (upper), immunoprecipitated with α-HA antibody (for Netrin-1) or α-Flag antibody (for EGFR/ECD) and detected using α-HA antibody (for Netrin-1) or α-Flag antibody (for EGFR/ECD) (lower). E: Dox-inducible Netrin-1-HA-overexpressing HepG2-NTCP-YFP cells were transfected with plasmids encoding wild-type EGFR-Flag (upper) or EGFR-ECD-Flag (lower). After 48 h of incubation with 25 ng/mL Dox, the cells were fixed and stained with anti-HA-tag (green) and anti-Flag-tag (red) antibodies, and DAPI (blue) for nuclear staining. The samples were then observed by confocal microscopy. Colocalization of Netrin-1 with EGFR is indicated by yellow fluorescence. F: Dox-inducible wild-type Netrin-1-HA-overexpressing HepG2-NTCP-YFP cells were transfected with wild-type EGFR-Flag overexpression plasmids and cultured with 0, 1, 5, or 25 ng/mL Dox for 48 h. Immunoprecipitation was performed using α-Flag antibody (for EGFR) and detected by α-GFP antibody (for NTCP), α-HA antibody (for Netrin-1), or α-Flag antibody (for EGFR). G: A model depicting the interaction between Netrin-1 and EGFR, and its inhibitory effect on the binding of NTCP to EGFR and HBV-NTCP internalization. Data are the mean ± standard error of the mean (n = 3). Statistical analysis was performed using two-way ANOVA with Tukey's test. ****$p < 0.0001$, ***$p < 0.001$, *$p < 0.05$, ns = not significant.

Suppression of HBV internalization by Netrin-1 was also evaluated using the preS1-probe. After preS1-probe attachment, the cells were incubated at 37°C to facilitate preS1-probe uptake, and the internalized preS1-probe signal was detected over 6 h (Fig 5B and 5C). A time-dependent increase in the internalized preS1-probe signal was observed until 4 h (Fig 5B). Dox-induced (25 ng/mL) Netrin-1 overexpression significantly decreased preS1-probe internalization (Fig 5B). Confocal immunofluorescent microscopy observation clearly demonstrated decreased preS1-probe internalization following Netrin-1 overexpression together with the decreased internalization of NTCP (Fig 5C). PreS1-probe internalization was also evaluated in the presence of EGF (S9 Fig). EGF treatment significantly increased preS1-probe internalization, and Netrin-1 overexpression significantly decreased preS1-probe internalization in the presence of EGF (S9 Fig). Thus, Netrin-1 suppresses HBV internalization independently of HBV attachment.

To explore the molecular basis of the effect of Netrin-1 on HBV internalization, we examined the interaction of Netrin-1, NTCP, and EGFR. Interestingly, Co-IP analysis showed that Netrin-1 did not bind to NTCP, whereas it did bind to EGFR (Fig 5D). We confirmed that Netrin-1 could bind to the extracellular domain (ECD) of EGFR (Fig 5D). Immunofluorescent staining of Netrin-1 and EGFR showed the colocalization of these two proteins on the cell surface and in the cytoplasm (Fig 5E).

During HBV internalization, the HBV-NTCP complex interacts with EGFR and is subsequently internalized into the cell along with EGFR [20,21]. Therefore, we examined whether Netrin-1 interferes with the binding of NTCP and EGFR. Co-IP analysis showed that Dox-induced Netrin-1 overexpression decreased the amount of NTCP pulled down by EGFR in a dose-dependent fashion (Fig 5F). Thus, Netrin-1 could impair NTCP-EGFR complex formation and suppress EGFR-mediated HBV internalization (Fig 5G).

## Netrin-1 impairs the phosphorylation and dimerization of EGFR, critical processes for HBV internalization

EGFR consists of an ECD, an α-helix transmembrane domain, an intracellular juxtamembrane domain, a tyrosine kinase domain, and a C-terminal regulatory domain (Fig 6A). Activated EGFR undergoes dimerization, followed by autophosphorylation at sites within the tyrosine kinase domain, ultimately triggering downstream signaling pathways, including PI3K-AKT, JAK-STAT, and Ras-MAPK, as well as its own endocytosis [22,23]. For the internalization of EGFR, the carboxyl-terminal region of EGFR is important and C-terminal truncation or mutations at Y1068, Y1148, and Y1173 significantly reduce EGFR internalization [24].

We examined whether Netrin-1 affects EGFR activation. Serum-starved HepG2-NTCP-YFP cells were stimulated with EGF in the presence of various concentrations of recombinant Netrin-1. Recombinant Netrin-1 blocked the PI3K-AKT pathway without affecting the Ras-MAPK pathway (Fig 6B). This effect was further confirmed using Dox-inducible Netrin-1-overexpressing HepG2-NTCP-YFP cells (Fig 6C). We evaluated the phosphorylation status of several tyrosine phosphorylation sites within EGFR, such as Y845, which serves as a docking site for downstream adaptor proteins to mediate cellular signaling pathways, or Y1068 and Y1173, which are phosphorylation residues in the C-terminal region that are essential for receptor internalization (Fig 6A) [25–27]. Netrin-1 reduced the levels of these phosphorylated residues (p-EGFR-845, p-EGFR-1068, p-EGFR-1173, and p-EGFR-total) in a dose-dependent manner (Fig 6D). The reduction of EGFR phosphorylation (p-EGFR-1173 and p-EGFR-total) was also observed in recombinant Netrin-1-treated HepG2-NTCP-YFP cells (Fig 6E).

We evaluated EGFR internalization in Dox-inducible Netrin-1-overexpressing HepG2-NTCP-YFP cells (S10 Fig). EGF stimulation substantially increased the amount of internalized EGFR that was located at the submembrane space as detected by immunofluorescent staining, while Dox-induced Netrin-1 significantly reduced the amount of internalized EGFR (S10 Fig).

We further examined whether HBS mutations in the V and C domains of Netrin-1 had any effect on the phosphorylation of these residues. The HBS mutant of Netrin-1 interacted with EGFR (Fig 6F) and suppressed the levels of these phosphorylated residues (p-EGFR-845, p-EGFR-1068, p-EGFR-1173, and p-EGFR-total), similar to wild-type Netrin-1 (Fig 6G).

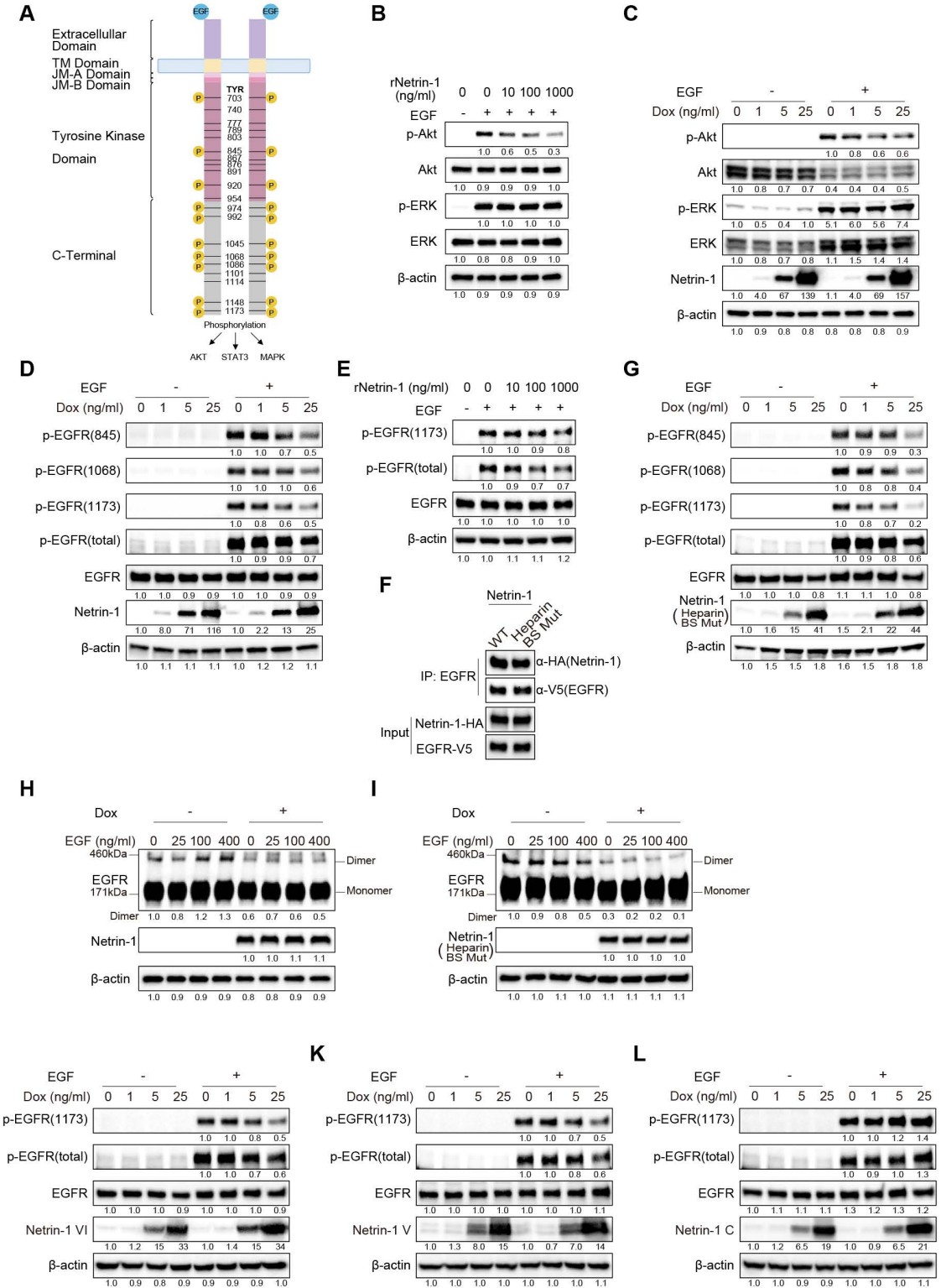

**Fig 6. Netrin-1 impairs the phosphorylation and dimerization of EGFR, critical processes for HBV internalization.** A: Diagrammatic overview of the domains of EGFR. B: Serum-starved HepG2-NTCP-YFP cells were stimulated with 100 ng/mL EGF for 15 min in the presence of the indicated concentrations of recombinant Netrin-1 (rNetrin-1). The cells were harvested for immunoblot analysis using antibodies against phosphorylated Akt, total

Akt, phosphorylated ERK, and total ERK. β-Actin was used as a loading control. C: Dox-inducible Netrin-1-overexpressing HepG2-NTCP-YFP cells were cultured with 0, 1, 5, or 25 ng/mL Dox for 48 h, and starved for 24 h followed by stimulation with 0 or 100 ng/mL EGF for 15 min. The cells were harvested for immunoblot analysis as described in panel B. D: Samples under the same conditions as in panel C were subjected to immunoblot analysis using antibodies to phosphorylated EGFR tyrosine residues (Y845, Y1068, and Y1173), as well as total phosphorylated EGFR and total EGFR. E: Serum-starved HepG2-NTCP-YFP cells were pre-treated with 0, 10, 100, or 1,000 ng/mL rNetrin-1 for 2 h, followed by stimulation with 100 ng/mL EGF for 15 min in the continued presence of the indicated concentrations of rNetrin-1. The cells were harvested for immunoblot analysis using antibodies to phosphorylated EGFR tyrosine residues (Y1173), total phosphorylated EGFR, and total EGFR. β-Actin was used as a loading control. F: Wild-type EGFR-V5 over-expression plasmids were transfected into Dox-inducible wild-type Netrin-1- or heparin-binding-deficient mutant Netrin-1-HA-overexpressing HepG2-NTCP-YFP cells. After 48 h of culture with 25 ng/mL Dox, the cells were subjected to immunoprecipitation using α-V5 antibody (for EGFR) and detected by α-HA antibody (for Netrin-1) or α-V5 antibody (for EGFR). G: Dox-inducible heparin-binding-deficient mutant Netrin-1-overexpressing HepG2-NTCP-YFP cells were prepared and analyzed as described in panel D. H and I: EGFR dimerization was assessed in Dox-inducible wild-type Netrin-1 (H) or heparin-binding-deficient mutant Netrin-1 (I) overexpressing HepG2-NTCP-YFP cells, cultured in the presence or absence of 25 ng/mL Dox. After 24 h of starvation, the cells were stimulated with the indicated concentrations of EGF at 4°C for 30 min. BS$^3$ (3 mM, final concentration) was added and the cells were incubated at 4°C for 2 h to crosslink EGFR dimers. The samples were analyzed by sodium dodecyl sulfate-polyacrylamide gel electrophoresis and western blotting. J, K, and L: Dox-inducible truncated Netrin-1 variant-overexpressing HepG2-NTCP-YFP cells were prepared, and phosphorylated EGFR (Y1173 and total phosphorylated EGFR) was analyzed.

EGF binding induces structural changes in the ECD of EGFR, which subsequently triggers the dimerization of the intracellular domain [28]. An assay using bis(sulfosuccinimidyl) suberate (BS$^3$) cross-linking was performed to analyze EGFR dimerization [29]. EGF-stimulated and Dox (25 ng/mL)-induced Netrin-1-overexpressing HepG2-NTCP-YFP cells were treated with BS$^3$ to facilitate EGFR cross-linking. Wild-type and HBS-mutant Netrin-1 both inhibited EGFR dimerization (Fig 6H and 6I), suggesting that the interaction of Netrin-1 and EGFR mediates structural changes of EGFR to inhibit its dimerization, and these structural changes are irrelevant for heparin binding.

To explore which domains of Netrin-1 are responsible for the inhibition of HBV internalization, we examined the domains of Netrin-1 for binding to the ECD of EGFR. Co-IP analysis showed that Netrin-1 bound to EGFR via multiple domains (VI, V, and C) (S11 Fig), similar to its binding to the UNC5 receptor [17] and LIPG. The influence of each domain of Netrin-1 on the phosphorylation of EGFR was analyzed. Interestingly, the VI and V domains of Netrin-1 exhibited a decreasing effect on the levels of C-terminal p-EGFR (1173) and p-EGFR (total), while the C domain did not (Fig 6J–6L). Thus, Netrin-1 inhibits HBV internalization and attachment by different modes of action. The VI and V domains are functionally important for the inhibition of HBV internalization, while the inhibition of HBV internalization is independent of heparin binding.

## Recombinant Netrin-1 inhibits HBV infection in humanized hepatocyte chimeric (TK-NOG) mice

To further investigate the *in vivo* role of Netrin-1 in HBV infection, recombinant Netrin-1 protein was administered to TK-NOG mice [30], followed by assessment of HBV infection. Flag-tagged full-length recombinant Netrin-1 (rNetrin-1) was purified from HEK293F cells. An osmotic pump containing either rNetrin-1 or PBS (as a negative control) was subcutaneously implanted into the mice. The pump delivered rNetrin-1 continuously at a rate of 5 μg in 6 μL per day for 4 weeks, and the mice were inoculated with HBV three days after pump implantation. Serum HBV DNA levels were subsequently monitored for five weeks until sacrifice. In addition, because the inhibitory effect of rNetrin-1 observed in the second week was weaker than that in the first week, additional boost of intravenous injections of rNetrin-1 (5ug/time, three times per week until the fifth week) via the tail vein were initiated from the second week (Fig 7A). Consequently, no obvious side effects were observed, and body weight as well as serum aspartate aminotransferase (AST) levels showed no significant differences between the rNetrin-1-treated and control groups (Fig 7B and 7C). Serum HBV DNA levels were significantly decreased in the rNetrin-1-treated group, with an approximately 2-log reduction in the fifth week (Fig 7D). Consistently, hepatic HBV DNA and cccDNA levels were also decreased in the rNetrin-1-treated mice (Fig 7E and 7F). Western blotting analysis confirmed the presence of Flag-tagged rNetrin-1 in liver tissue of the rNetrin-1-treated group (Fig 7G). However, no significant increase in serum Netrin-1 levels were observed in blood samples collected during rNetrin-1 administration

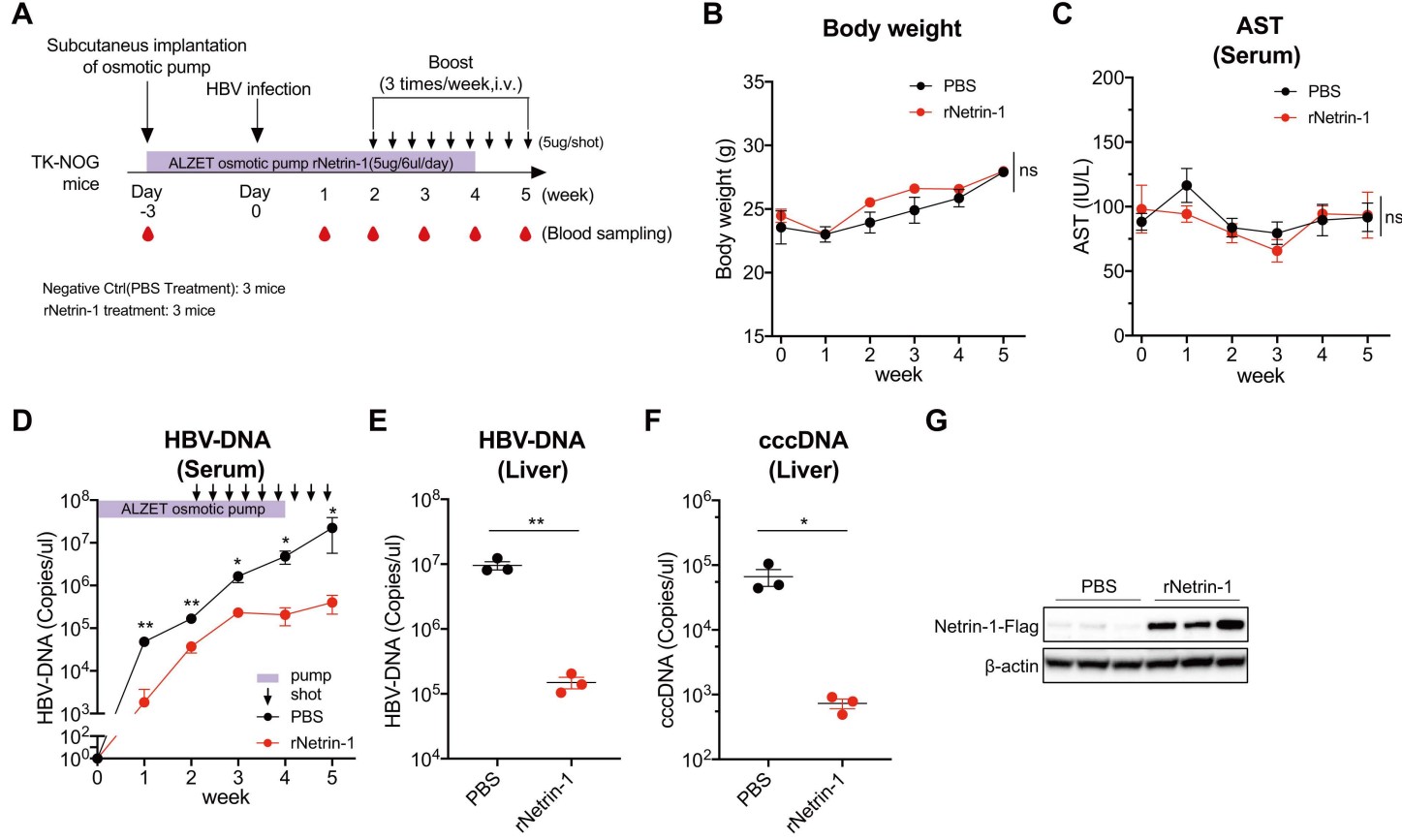

**Fig 7. Recombinant Netrin-1 inhibits HBV infection in humanized hepatocyte chimeric (TK-NOG) mice.** A: Schematic timeline of rNetrin-1 treatment and HBV infection in TK-NOG mice (related to B–G). The ALZET osmotic pump containing rNetrin-1 (delivering 5 μg in 6 μL per day) or PBS as a control was implanted subcutaneously (n = 3 per group). Three days later, mice were inoculated with HBV (genotype D; 5.0 × 10⁶ GEq in 100 μL) via tail vein injection. From the second week onward, rNetrin-1 (5 μg in 100 μL PBS per dose) was additionally administered three times per week by tail vein injection. Blood samples were collected prior to pump implantation and once weekly after HBV inoculation until sacrifice, five weeks post-inoculation. B and C: Body weight (B) and mouse serum aspartate aminotransferase (AST) levels (C) of mice treated with rNetrin-1 or PBS, monitored at the indicated time points. D: qPCR analysis of mouse serum HBV-DNA at the indicated time points. E and F: qPCR analysis of HBV-DNA (E) and cccDNA (F) in liver tissues of mice at the experimental endpoint. G: Western blotting analysis of liver tissues of mice at the experimental endpoint. Netrin-1 was detected using an α-Flag antibody. The mean ± standard error of the mean of three mice in each group is indicated. Statistical testing was performed using a two-tailed unpaired $t$-test. ****$p < 0.0001$, ***$p < 0.001$, *$p < 0.05$, ns = not significant.

(S12A Fig). To clarify this observation, we assessed the pharmacokinetics by evaluating the effect of a single intravenous bolus of rNetrin-1 (5 μg in 100 μL PBS). After administration, serum Netrin-1 reached a maximum concentration of approximately 800 pg/mL at 4 hours and subsequently declined to baseline levels within 24 hours (S12B Fig).

## Discussion

Our previous study suggested that LIPG promotes HBV entry by enhancing viral binding to HSPGs or NTCP [6]. Additionally, *in vitro* virus screening identified Netrin-1 as a potential LIPG-interacting protein, and synthetic Netrin-1 peptides were shown to inhibit HBV infection [7]. Building on these findings, we investigated the role of Netrin-1 in the HBV entry process.

In HBV infection assays using primary human hepatocytes (PXB cells), Netrin-1 exhibited the strongest inhibitory effect among known LIPG-binding partners (Fig 1C and 1D, S1 Fig). We showed that Netrin-1 significantly suppressed viral

entry, affecting the attachment and internalization steps (Fig 1K, 1L, 1O, and 1P). HBV attachment begins with low-affinity binding to HSPGs, followed by a high-affinity interaction with NTCP [4]. PreS1 binding assays and enzymatic removal of HSPGs demonstrated that Netrin-1 specifically impairs the HSPG-dependent attachment step (Fig 2B and 2C). Classic Netrin-1 receptors such as UNC5B and neogenin were not involved in this process (S5 Fig).

Despite the appropriate expression of Netrin-1 in PXB cells, it was expressed at a low level in tumor cells; therefore, we established Dox-inducible Netrin-1-overexpressing HepG2-NTCP-YFP cells (Fig 1M). Transwell experiments clearly demonstrated that Dox-induced Netrin-1 was secreted into the culture medium (S2A Fig) and accumulated on the cell membrane of non-Netrin-1-overexpressing HepG2-NTCP-YFP cells (Fig 1R). This transferred Netrin-1 inhibited HBV attachment and internalization (Fig 1S and 1T). Thus, the results clearly show that Netrin-1 regulates HBV entry in a *trans*-acting manner.

For the development of Netrin-1-based anti-HBV drug development, we explored the precise mechanism of interaction between Netrin-1 and LIPG. Netrin-1 binds to HSPGs via basic regions of the HBSs in the V[15] and C[16] domains (Figs 2D, 3A, and 3B). LIPG also anchors to HSPGs via HBSs and has been proposed to form a bridge between HBV and the hepatocyte surface [6,18]. Co-localization and binding assays confirmed a physical interaction between Netrin-1 and LIPG (Figs 2E, S6 and S7). Notably, Netrin-1 did not bind directly to HBV surface proteins (Fig 2E), implying an indirect inhibitory mechanism.

Domain mapping revealed that the V and C domains of Netrin-1 were functionally important for suppressing HBV attachment (Fig 2H). In more detail, the HBS in the V domain interacted with the HBS in the N-terminal half of LIPG (LIPG-NTD), while the HBS in the C domain interacted with the HBS in the C-terminal half of LIPG (LIPG-CTD), and mutations of the HBS in the V or C domain abolished these interactions (Fig 3D–3I). It could be speculated that the interaction between the HBSs of Netrin-1 and LIPG could be mediated through free heparin present in the culture medium. The linear structure of heparin with multiple binding sites can bind two or more proteins simultaneously, allowing for oligomerization or aggregation [31]. In this study, we performed functional analysis of individual domains (VI, V, and C) (Fig 2G and 2H); however, we were unable to determine whether these proteins are present in physiological conditions. Nevertheless, truncated Netrin-1 (VI-V) has been reported to play significant biological functions in humans [32]. It was also interesting that the individual domains showed greater stability than full-length Netrin-1 in the culture supernatant (Fig 2G). The truncated conformational structures of these proteins might promote their oligomerization or aggregation.

Our results showed that the V and C domains of Netrin-1 suppressed HBV attachment in different manners. The V domain suppressed the binding of LIPG to HSPGs by competing out LIPG from HSPGs (Fig 4J) or masking the HBS of LIPG (Fig 3J), while the C domain suppressed the binding of LIPG to LHBs by competing out LHBs from LIPG (Fig 4G). A double mutant of the HBSs in the V and C domains lost the ability to suppress the attachment of HBV induced by Netrin-1 (Fig 4I). Thus, Netrin-1 inhibited HBV attachment by three possible mechanisms: (1) competing with LIPG for HSPG binding (Fig 4K-b); (2) masking the HBS on LIPG (Fig 4K-c); and (3) preventing LIPG from binding to LHBs (Fig 4K-d).

In addition to blocking viral attachment, Netrin-1 suppressed HBV internalization (Figs 1P, 1T, and 5A–5C–5C). EGFR, a known HBV entry cofactor [20], is involved in this process. Netrin-1 did not bind to NTCP, but interacted with the ECD of EGFR (Fig 5D and 5E), disrupting the EGFR-NTCP association (Fig 5F and 5G). EGFR activation involves autophosphorylation and dimerization, triggering pathways such as PI3K-AKT and receptor endocytosis [22,23]. The C-terminal region of EGFR is crucial for its internalization, and mutations at Y1068, Y1148, or Y1173 are known to impair this process [24]. Netrin-1 reduced phosphorylation at multiple EGFR sites (Y845, Y1068, and Y1173), inhibited EGFR dimerization, and suppressed downstream signaling (Fig 6D–6I). These results were partially confirmed by adding recombinant Netrin-1 to the culture medium (Fig 6E). In addition, EGF-induced internalizations of the preS1-probe and EGFR were clearly reduced by Dox-induced Netrin-1-overexpression (S9 and S10 Figs). These effects were observed even with a heparin-binding-deficient mutant, indicating that Netrin-1 modulates EGFR independently of HSPG binding (Fig 6G and 6I). Domain mapping suggested that the VI and V domains are primarily responsible for EGFR suppression (Fig 6J–6L).

Our *in vivo* studies demonstrate that exogenously administered rNetrin-1 successfully localized to liver tissue and effectively reduced HBV infection in TK-NOG mice, as evidenced by decreased serum and hepatic viral DNA levels (Fig 7D–7G). In the pharmacokinetic analysis, serum Netrin-1 reached its maximum concentration 4 hours after intravenous bolus administration, which appears later than expected (S12B Fig). It has been previously demonstrated that endothelial adsorption can result in a delayed attainment of the maximum plasma concentration for biopharmaceuticals following intravenous infusion [33,34]. As a pleiotropic ligand, Netrin-1 first interacts with cell-surface HSPGs before specifically engaging its cognate receptors, which mediate diverse biological processes including angiogenesis, inflammation, tumorigenesis, and tissue regeneration [8]. Transient interactions with HSPGs on the surfaces of endothelial or other peripheral cells prior to reaching systemic circulation may contribute to the observed delay in peak serum concentration. Despite continuous delivery of rNetrin-1 via an osmotic pump and supplemental intravenous injections, serum Netrin-1 levels in the samples did not exhibit any corresponding increase (S12A Fig). The pharmacokinetic analysis revealed that rNetrin-1 is rapidly cleared from the circulation, with serum levels returning to baseline within 24 hours after a single intravenous bolus (S12B Fig). Notably, blood samples for weeks 3–5, corresponding to the period of supplemental intravenous injections, were collected 48 h after the injections. Although it remains uncertain whether rNetrin-1 retains its protein stability within the pump, our pharmacokinetic analysis demonstrated that rNetrin-1 exhibits a short systemic half-life *in vivo*, likely attributable to rapid turnover and/or efficient uptake by peripheral tissues such as the endothelium and liver. These pharmacokinetic properties likely account for the absence of detectable increases in serum Netrin-1 levels following exogenous rNetrin-1 administration in this study.

Netrin-1 is a secreted laminin-related protein and plays a pivotal role in neural development and guiding axons [35]. However, its expression is not limited to the nervous system and it is also involved in processes such as angiogenesis, inflammation, tumorigenesis, and tissue regeneration. The expression of Netrin-1 has not been fully evaluated in hepatocytes. Therefore, we examined the changes in expression of Netrin-1 and LIPG in hepatocytes and culture medium following HBV infection (S4 Fig). Netrin-1 was expressed in the cell lysates and culture medium of PXB cells (S4 Fig), and its concentrations in the culture medium (30–90 pg/mL) were comparable to those reported in healthy human serum (mean 130 pg/mL) [36]. Expression of Netrin-1 decreased gradually in normal hepatocytes (PXB cells) (S4A, S4C, and S4D Figs) and tumor cells (HepG2-NTCPsec+) (S4E Fig) over the culture period. The differentiation of PXB cells starts from day 3 to day 5 and they maintain the function of hepatocytes from day 7 to day 10, which then decreases after day 14 [37]. Therefore, as Netrin-1 is prone to be expressed in progenitor cells [35], the general repression of Netrin-1 expression during cell culture could be explained by the progression of hepatocyte differentiation. In addition, interestingly, Netrin-1 was rather repressed in HBV-infected HepG2-NTCPsec+ cells (S4E Fig) and PXB cells at the late phase of HBV infection (day 8 to day 12) (S4D Fig). These findings were in sharp contrast to those reported by Plissonnier et al. [38], who stated that Netrin-1 expression was induced by the NS5A protein of HCV and promoted HCV entry [38]. The reasons for the different effects of Netrin-1 on HCV and HBV are unknown. Plissonnier et al. also reported that Netrin-1 changed the HCV virion and activated EGFR, thereby enhancing HCV infection. All of their assays were performed in the presence of HCV infection [38]. In contrast, we showed that Netrin-1 disrupted the association of EGFR and NTCP (Fig 5F and 5G) and inactivated EGFR in the absence of HBV infection (Fig 6). If Netrin-1 has different effects on HCV and HBV, in the case of HCV/HBV coinfection, the increased expression of Netrin-1 induced by HCV should return to normal by the eradication of HCV, which could potentially trigger HBV reactivation, as occasionally observed in direct-acting antiviral treatment [39].

Although the underlying mechanisms for the reduction of Netrin-1 levels by HBV infection are unknown, its expression could be regulated by multiple factors such as cell differentiation, inflammation, and apoptosis. Moreover, in the clinical setting of HBV infection, immune cells could also be involved in the regulation of Netrin-1 levels. Further study should be performed to clarify the relationship between HBV and the regulation of Netrin-1 expression *in vitro* and *in vivo*.

Another limitation of this study is that our protein-protein interaction analyses were mainly performed using a cell transfection assay. Netrin-1 and LIPG are secreted into the culture medium and attach to HSPGs on the cell membrane, and

the levels of these proteins in the medium and cell membrane are similar. All of the constructs used in this study (various truncations of Netrin-1 and LIPG) were cloned from the original sequences, as described in the Materials and Methods, so we expected that the expressed proteins would be secreted and attach to the cell membrane. We assume that our results reflect the proteins on the cell membrane or in the medium adjacent to the cell membrane (Fig 2G). However, the possibility remains that our findings may reflect intracellular proteins, although the Transwell experiments clearly showed that secreted Netrin-1 exerts an inhibitory effect on HBV infection in a *trans*-acting manner. Moreover, our results do not reflect the events in cell-free medium. Further analysis should be carried out to examine whether our data from the culture medium are valid by collecting secreted proteins individually.

In conclusion, our findings reveal a novel antiviral mechanism of Netrin-1 that targets multiple steps in HBV entry. The entry step of HBV is critical for the formation of cccDNA. Although cccDNA replenishment is limited in *in vitro* and *in vivo* experiments [40,41], to maintain HBV infection in patients with chronic hepatitis B, new infections of uninfected hepatocytes are necessary because hepatocyte regeneration is induced by passive senescence-induced loss or inflammation--induced active cell death. Therefore, blocking the entry step of HBV should be an important strategy to eliminate cccDNA in combination with other therapies. To that aim, optimal delivery system of rNetrin-1 should be developed and future studies investigating the structural basis of the interactions of Netrin-1 with LIPG and EGFR will be critical to unlocking its full antiviral potential.

## Materials and methods

### Ethics statements

All animal experiments were approved by the Ethics Committee for the Care and Use of Laboratory Animals at the Takara-Machi Campus of Kanazawa University, Japan and were carried out in compliance with the ARRIVE guidelines 2.0. All experiments were performed in accordance with relevant guidelines and regulations.

### Cell culture

HepG2-NTCP-YFP, Huh7-NTCP-YFP, HepG2, HepG2.2.15, HepAD38, and HepG2-NTCPsec+ cells were maintained in Dulbecco's modified Eagle's medium/F-12, GlutaMAX supplement (Gibco, Carlsbad, CA) containing 10% fetal bovine serum (Thermo Fisher Scientific, Waltham, MA), 10 mM HEPES (Dojindo, Kumamoto, Japan), 5 µg/mL insulin (Sigma-Aldrich, St. Louis, MO), and 1% penicillin/streptomycin (Thermo Fisher Scientific). HepG2.2.15 and HepAD38 cells also required 400 mg/mL G418. PXB cells were obtained from PhoenixBio (Hiroshima, Japan) and cultured with the specifically designed dHCGM (PhoenixBio).

### Establishment of the HepG2-NTCP-YFP cell line

The HepG2-NTCP-YFP cell line was established in the same manner as the Huh7-NTCP-YFP cells, as described previously [6]. The NTCP-YFP expression vector, pENNTCPY, was constructed by fusing YFP DNA from pEYFP-N1 (Takara Bio USA, San Jose, CA) to the C-terminus of NTCP cDNA, which was synthesized from human liver total RNA (Takara Bio USA). This fusion construct was inserted downstream of the CMV/chicken β-actin promoter in a pEB Multi-Neo derivative vector (Fujifilm Wako Pure Chemical Corp., Osaka, Japan). HepG2-NTCP-YFP cells were generated by transfecting HepG2 cells with the pENNTCPY plasmid.

### Establishment of Dox-inducible HepG2-NTCP-YFP cells expressing Netrin-1 or LIPG

Open reading frames of the wild-type and truncated forms of Netrin-1 and LIPG (with the signal sequence retained to ensure proper secretion) were amplified using the Netrin-1 cDNA plasmid (RC218429; OriGene, Rockville, MD) or LIPG cDNA plasmid (RC209248; OriGene) as templates. The resulting DNA fragments were cloned into the pLVX-TetOne-Puro

or pLVX-TetOne-Hygro vector (Takara Bio USA). Dox-inducible stable HepG2-NTCP-YFP cell lines were then generated using the Lenti-X Tet-One Inducible Expression System (Takara Bio USA).

## shRNA-mediated knockdown and mRNA expression analysis by real-time qPCR

shRNAs targeting the five binding partners of LIPG (target sequences listed in S1 Table) (Sigma-Aldrich) were cloned into a lentiviral-based pLKO.1-puro expression vector. Lentiviruses were generated by using MISSION Lentiviral Packaging Mix (Sigma-Aldrich). A PLKO.1 puro non-target shRNA control vector was used to generate the negative control virus. PXB cells were transduced with lentiviruses, and fresh medium was replaced after 16 h. The cells were then maintained for 5 days prior to HBV infection. Total RNA was extracted from the cells using an RNAqueous-Micro Total RNA Isolation Kit (Thermo Fisher Scientific). Subsequently, 100 ng total RNA was reverse-transcribed into cDNA using a High-Capacity cDNA Reverse Transcription Kit (Applied Biosystems). qPCR was performed with gene-specific primers (listed in S1 Table) and SYBR Green PCR Master Mix (Applied Biosystems) using the 7500 Real-Time PCR System (Applied Biosystems).

## DNA transfection

Open reading frames from cDNA plasmids for Netrin-1 (RC218429; OriGene), LIPG (RC209248; OriGene), wild-type EGFR (FXC09683; Kazusa DNA Research Institute, Chiba, Japan), ECD of EGFR (RC217223; OriGene), and truncated variants of Netrin-1 or LIPG (with the signal sequence retained) were inserted into a digested pcDNA3.1(+) mammalian expression vector (V79020; Invitrogen, Carlsbad, CA). The reconstructed plasmids were transfected into cells with FuGENE HD Transfection Reagent (Promega, Madison, WI).

## HBV infection

HBV stock was derived from HepAD38 cell (genotype D) culture supernatant or obtained from PhoenixBio (genotype C; Hiroshima, Japan). To prepare the HepAD38 cell-derived HBV, the supernatant passed over 0.45-µm VWR filter units was concentrated with a PEG Virus Precipitation Kit (BioVision, Milpitas, CA, USA). Purified HBV was quantified by HBV-DNA qPCR. For HBV infection, PXB or HepG2-NTCPsec+ cells were inoculated with HBV (genotype C; PhoenixBio) at 5 or 10 genome equivalents (GEq)/cell or HepAD38 cell-derived HBV at 250,000 GEq/cell for 16 h in medium containing 4% PEG 8000 (Merck, Kenilworth, NJ). After washing with PBS, the cells were cultured according to the schedule shown in the figure legend.

## HBV attachment assay

To allow HBV to bind to the cell surface, HepG2-NTCP-YFP or Huh7-NTCP-YFP cells were incubated with 50,000 GEq/cell cold HBV at 4°C for 1.5 h. Free virus was washed out and the cells were subjected to DNA extraction. HBV attachment was assessed by qPCR targeting cell-bound HBV DNA.

## HBV internalization assay

HepG2-NTCP-YFP or Huh7-NTCP-YFP cells were incubated with 50,000 GEq/cell cold HBV at 4°C for 1.5 h. Free virus was washed out and the cells were shifted to 37°C for 6 h. After washing with PBS, the cells were trypsinized with 0.25% trypsin for 10 min to remove cell surface-attached HBV. The cells were harvested for DNA extraction, and internalized HBV was quantified by qPCR.

## HBV replication assay

HepG2.2.15 cells were either transfected with the Netrin-1 expression plasmid or incubated with recombinant Netrin-1 protein. After 5 days of culture, the extracellular HBV DNA in the culture supernatant was collected and quantified using an HBV Quantitative Assay Kit (KUBIX, Hakusan, Japan).

## PreS1 binding assay

This protocol was described previously [6]. The preS1-probe, an N-terminally myristoylated and C-terminally TAMRA-conjugated peptide, sharing 2–48 amino acids of the HBV preS1 region was used. In brief, 80 nM preS1-probe was used to treat cells at 37°C for 1.5 h. After incubation, the cells were fixed with 4% paraformaldehyde following by DAPI staining. Finally, fluorescence imaging was performed using the CellVoyager CQ1 confocal microscopy system (Yokogawa Electric Corporation, Tokyo, Japan).

## PreS1 internalization assay

The cells were incubated with 80 nM preS1-probe at 4°C for 1.5 h, followed by incubation at 37°C. Afterward, the cells were fixed with 4% paraformaldehyde and stained with DAPI. Internalization of the preS1-probe was observed using the CellVoyager CQ1 imaging system (Yokogawa Electric Corporation), and the data were analyzed with ImageJ software (National Institutes of Health, Bethesda, MD).

## Indirect immunofluorescence analysis

Samples fixed in 4% paraformaldehyde were permeabilized with 0.3% Triton X-100. The samples were incubated with antibodies against Myc-Tag (2272) or HA-Tag (3724; Cell Signaling Technology [CST], Danvers, MA), followed by treatment with Alexa Fluor 555 (A32732; Invitrogen) or Alexa Fluor 660 (A-21074; Invitrogen) conjugated secondary antibodies. Alternatively, the samples were stained using anti-HA-tag mAb-Alexa Fluor 488 (M180-A48) and anti-DDDDK-tag mAb-Alexa Fluor 594 (M185-A59).

## Heparanase treatment

Recombinant Human Active Heparanase (7570-GH-005; R&D Systems, Minneapolis, MN) at a final concentration of 0.5 µg/mL was added to the cells followed by incubation at 37°C for 1.5 h. After incubation, the medium was removed and the cells were used in the next step of the HBV attachment assay.

## Extraction of HBV DNA and quantitative real-time PCR

HBV DNA was extracted from the cells using a DNeasy Blood & Tissue Kit (QIAGEN, Valencia, CA). HBV DNA quantitative real-time PCR was performed using a forward primer (5′-ACTCACCAACCTCCTGTCCT-3′), reverse primer (5′-GACAAACGGGCAACATACCT-3′), and specific HBV DNA probe (5′-FAM/TATCGCTGG/ZEN/ATGTGTCTGCGGC-GT/3IBFQ-3′). TaqMan Fast Advanced Master Mix (Applied Biosystems) was used and RT-PCR was performed on the 7500 Real Time PCR System (Applied Biosystems).

## Co-IP analysis

Cells transfected with plasmids or reaction mixtures from the TNT Quick Coupled Transcription/Translation System (Promega) were subjected to Co-IP assays. Immunoprecipitation was performed using a Dynabeads Co-Immunoprecipitation Kit (Invitrogen). The beads were pre-incubated with an anti-DDDDK-tag mAb (M185-3L; Medical and Biological Lab. Co., Ltd. [MBL], Aichi, Japan), anti-HA-tag mAb (M180-3; MBL), anti-V5-tag mAb (M215-3; MBL), mouse IgG2a (isotype control) (M076-3; MBL), or mouse IgG2b (isotype control) (M077-3; MBL).

## Cross-linking of EGFR

Cells starved for 24 h were stimulated with 0–400 ng/mL EGF (PeproTech, Rocky Hill, NJ) at 4°C for 30 min. After washing three times with cold PBS, the cells were subjected to a final concentration of 3 mM BS$^3$ (21580; Thermo Fisher Scientific) to allow cross-linking. After incubation at 4°C for 2 h, a quenching solution (10 mM Tris, pH 7.5) was used to terminate

the reaction for 15 min at 4°C. Subsequently, cold PBS-washed samples were lysed using Pierce IP lysis buffer (87788; Thermo Fisher Scientific) containing complete Protease Inhibitor Cocktail and PhosSTOP (Roche Applied Science, Pleasanton, CA). EGFR dimerization was analyzed by sodium dodecyl sulfate-polyacrylamide gel electrophoresis and western blotting.

### Western blotting

Western blotting was performed as described previously [6]. Culture supernatant samples were centrifuged at 1,700 rpm for 4 min to remove debris before analysis. The following antibodies were used: Netrin-1 (ab126729; Abcam, Cambridge, MA), LIPG (ab24447; Abcam), HBV-core (HBP-023–9; AUSTRAL Biologicals, San Ramon, CA), HBsAg (BCL-ABM2–02; Beacle, Inc., Kyoto, Japan), EGFR (4267; CST), p-EGFR(Tyr845) (2231; CST), p-EGFR(Tyr1068) (2234; CST), p-EGFR(Tyr1173) (4407; CST), p-Tyr(PY99) (sc-7020; Santa Cruz Biotechnology, Dallas, TX), Akt (4685; CST), p-AKT (9271; CST), p44/42 MAPK (Erk1/2) (9102; CST), p-p44/42 MAPK (Erk1/2) (9101; CST), GFP (mFX75; Wako), Na, K-ATPaseα1 (23565; CST), His-tag (12698; CST), anti-rabbit IgG (7074; CST), anti-mouse IgG (7076; CST), VeriBlot for IP Detection Reagent (HRP) (ab131366; Abcam), HA-tag (3724; CST) (M180-7; MBL), DDDDK-tag (PM020–7; MBL), V5-tag (M215-7; MBL), and β-actin (PM053–7; MBL).

### Southern blotting

Southern blotting was performed as described previously [6,42]. DNA from HBV-infected cells was extracted by the Hirt protein-free DNA extraction procedure. Then, DNA was electrophoresed on 1.2% agarose gels and transferred onto a Hybond-N+ membrane (GE Healthcare, Amersham, UK). The membrane was cross-linked by UV and hybridized with DIG-labeled single-stranded RNA probes. The hybridization signal was acquired by using the ChemiDoc Touch Imaging System (Bio-Rad, Hercules, CA).

### ELISA

The concentrations of Netrin-1 and LIPG in cell culture supernatants or mouse serum were measured using ELISA kits: CSB-E11899h (Cusabio, Wuhan, China) for human Netrin-1 and MBS702323 (MyBioSource, San Diego, CA, USA) for LIPG, following the manufacturers' instructions. Sample concentrations were calculated using the online Gaindata ELISA analysis tool [https://www.arigobio.com/elisa-calculator].

### Isolation of the membrane fraction

Cells were detached using 0.5 mM EDTA in PBS for 10 min at 37 °C. The harvested cells were subjected to membrane fractionation using a EzSubcell Extract Kit (WSE-7421; ATTO, Tokyo, Japan). The isolated membrane fraction was analyzed by sodium dodecyl sulfate-polyacrylamide gel electrophoresis and western blotting.

### Pull-down assay for biotinylated protein:protein interactions

A Pierce Pull-Down Biotinylated Protein:Protein Interaction Kit (21115; Thermo Fisher Scientific) was used to assess protein–protein interactions. A biotinylated peptide corresponding to the C-terminal HBS of LIPG (amino acids 331–340; sequence: NSKMYLKTRA) was synthesized and biotinylated at the N-terminus (custom-synthesized by GenScript, Piscataway, NJ). Recombinant large HBsAg (BCL-AGC-01; Beacle, Inc.) and carrier-free recombinant human Netrin-1-His (6419-N1-025/CF; R&D Systems) were used as prey proteins. The pull-down assay was performed according to the manufacturer's protocol. Following the interaction step, bound proteins were eluted from the streptavidin resin, mixed with sodium dodecyl sulfate sample buffer, and subjected to sodium dodecyl sulfate-polyacrylamide gel electrophoresis, followed by western blotting. Detection was carried out using an anti-His tag antibody for Netrin-1-His and an anti-HBsAg antibody for large HBsAg, as appropriate.

## Transswell co-culture assay

Dox-inducible Netrin-1-overexpressing HepG2-NTCP-YFP cells were seeded in Falcon cell culture inserts (0.4-μm pore, 353494, 353090; Corning, Inc., Corning, NY) placed in 12- or 6-well plates. Simultaneously, wild-type HepG2-NTCP-YFP cells were seeded in separate 12- or 6-well plates. On the following day, the inserts were transferred into wells containing wild-type HepG2-NTCP-YFP cells to establish a Transwell co-culture system. The cells were then treated with 0, 1, 5, or 25 ng/mL Dox for 72 h to induce Netrin-1 expression, followed by further analysis.

For HBV entry assays, the upper inserts containing Dox-inducible Netrin-1-overexpressing HepG2-NTCP-YFP cells were removed, and the wild-type HepG2-NTCP-YFP cells in the lower chamber were subjected to subsequent analysis. For western blotting, whole cell lysates from the upper insert cells, as well as whole cell lysates and membrane fractions from the lower chamber cells, were prepared and analyzed to detect Netrin-1-HA expression levels.

## Cell viability assay

A Cell Counting Kit-8 (341–08001; Wako) was used to determine cell viability.

## Mice

TK-NOG mice were provided by the Central Institute for Experimental Medicine and Life Science (Kawasaki, Japan). The ALZET osmotic pump (model 2004), pre-filled with rNetrin-1 in 200 μL PBS to deliver 5 μg in 6 μL per day, or PBS alone as a negative control, was subcutaneously implanted into the mice (n = 3 per group). Three days after pump implantation, mice were inoculated with HBV (genotype D; $5.0 \times 10^6$ GEq in 100 μL per mouse) derived from HepAD38 cells via tail vein injection. Beginning in the second week, rNetrin-1 (5 μg in 100 μL PBS per dose, three times per week to fifth week) was administered as boosts by intravenous injection via the tail vein. Serum aspartate aminotransferase levels were measured using a Transaminase CII-Test Wako kit (Fujifilm, Osaka, Japan).

## Recombinant human Netrin-1 purification

Recombinant human Netrin-1 was expressed in HEK293F cells using the Expi293 Expression System Kit (Thermo Fisher Scientific) and purified sequentially with a HiTrap Heparin HP column (Cytiva) and DDDDK-tagged Protein Purification Gel (MBL). The full-length Netrin-1 cDNA, including the native signal peptide, was cloned into pcDNA3.1(+) with a C-terminal 3 × DDDDDK tag. Transfected HEK293F cells were cultured for 6 days, lysed with Pierce IP Lysis Buffer (Thermo Fisher Scientific) containing protease inhibitors, and clarified by centrifugation (13,000 × g, 10 min, 4°C) and filtration (0.22 μm). The supernatant was buffer-exchanged to 10 mM sodium phosphate (pH 7.0) and applied to a HiTrap Heparin HP column. Bound proteins were eluted with 10 mM sodium phosphate containing 1.5 M NaCl (pH 7.0).

Netrin-1-containing fractions were pooled, concentrated, and further purified using DDDDK-tagged Protein Purification Gel. After washing with 30 mM Tris-HCl (pH 7.5), 150 mM NaCl, and 0.1% Triton X-100, the bound protein was eluted with 100 μg/mL 3 × FLAG peptide (ProteinArk) dissolved in 30 mM Tris-HCl (pH 7.5), 1 M NaCl, and 0.1% Triton X-100. The final product was buffer-exchanged to PBS, and its purity was confirmed by SDS-PAGE followed by Coomassie Brilliant Blue staining and Western blotting.

## Statistical analysis

All data were analyzed using Prism 8 software (GraphPad Software, Inc., San Diego, CA). At least three independent samples were included for each experimental condition. Statistical significance was assessed using unpaired *t*-tests or one-way or two-way analysis of variance (ANOVA) followed by Tukey's multiple comparisons test. *P*-values less than 0.05 were considered statistically significant.

## Supporting information

**S1 Fig. Functional evaluation of LIPG-binding partners in HBV infection.** A: Schematic representation of the experimental schedule for shRNA-lentivirus transduction and subsequent HBV infection in PXB cells. B: qPCR analysis of intracellular HBV DNA in PXB cells transduced with control shRNA or shRNAs targeting five LIPG-binding partners, including Netrin-1, measured at 17 days post-HBV infection. C: qPCR analysis of the mRNA levels of the five LIPG-binding partners in PXB cells at 5 days after shRNA-lentivirus transduction, confirming knockdown efficiency.
(TIF)

**S2 Fig. ELISA analysis of Netrin-1 and LIPG expression.** Culture supernatants from Dox-inducible Netrin-1-overexpressing HepG2-NTCP-YFP cells treated with 0, 1, 5, or 25 ng/mL Dox for 72 h (corresponding to Fig 1M–1T), were subjected to ELISAs to quantify Netrin-1 (A) and LIPG (B) levels.
(TIF)

**S3 Fig. Recombinant Netrin-1 exhibits an anti-HBV entry effect.** HepG2-NTCP-YFP cells were pre-treated with 0, 100, 1,000, or 1,500 ng/mL recombinant Netrin-1 (rNetrin-1) for 2 h, followed by western blotting analysis and HBV entry assays. Western blotting analysis of whole cell lysates from rNetrin-1-treated HepG2-NTCP-YFP cells was performed to detect rNetrin-1-His and β-actin (loading control). Netrin-1-His was detected using an anti-His tag antibody. Data shown represent the mean ± standard error of the mean from three independent experiments. Statistical testing was performed by two-way ANOVA with Tukey's multiple comparisons test. ****$p < 0.0001$, ***$p < 0.001$, *$p < 0.05$, ns = not significant.
(TIF)

**S4 Fig. Expression levels of Netrin-1 and LIPG in PXB cells and HepG2-derived HBV permissive cells with or without HBV infection.** A–C: PXB cells were infected with or without HBV and collected at early time points (days 3 and 6) for western blotting analysis. Intracellular hepatitis B core protein (HBc), Netrin-1, LIPG, and β-actin (loading control) were detected using the respective antibodies (A). Culture supernatants from the corresponding samples were subjected to ELISAs for the detection of LIPG (B) and Netrin-1 (C). D: PXB cells were infected with or without HBV and collected at later time points (days 8 and 12) post-infection for western blot analysis in an independent experiment. E and F: HepG2-NTCPsec+ cells were infected with or without HBV and collected at the indicated time points for western blotting analysis. Intracellular HBc, Netrin-1, LIPG, and β-actin (loading control) were detected as described in panel A. Culture supernatants from the corresponding samples were subjected to ELISAs for the detection of LIPG (F) and Netrin-1 (undetectable).
(TIF)

**S5 Fig. The typical receptors of Netrin-1 do not appear to be involved in HBV attachment and internalization.** A: Schematic representation of receptor interaction-deficient Netrin-1. B: Dox-inducible wild-type or receptor interaction-deficient Netrin-1-HA-overexpressing HepG2-NTCP-YFP cells were cultured in medium containing 0 or 25 ng/mL Dox for 72 h. The cells were then subjected to western blotting analysis (B), HBV attachment assay (C), or internalization assay (D). Data shown represent the mean ± standard error of the mean from three independent experiments. Statistical testing was performed by two-way ANOVA with Tukey's multiple comparisons test. ****$p < 0.0001$, ***$p < 0.001$, *$p < 0.05$, ns = not significant.
(TIF)

**S6 Fig. Colocalization of Netrin-1 and LIPG in the cell membrane.** HepG2 cells were co-transfected with Netrin-1-Myc and LIPG-HA overexpression plasmids. At 48 h after transfection, the cells were fixed and stained with the indicated antibodies. The cells were observed by super-resolution microscopy (scale bar, 2 μm). Colocalization of Netrin-1 with LIPG is indicated by yellow fluorescence.
(TIF)

**S7 Fig. Netrin-1-LIPG interaction in an *in vitro* translation system.** Overexpression plasmids encoding LIPG-V5 and either wild-type or truncated HA-tagged Netrin-1 variants were added to an *in vitro* translation reaction mixture. After incubation at 30°C for 1.5 h, the reaction mixtures, containing the synthesized recombinant proteins, were subjected to immunoprecipitation using α-V5 antibody (for LIPG) and detected by α-V5 antibody (for LIPG) or α-HA antibody (for Netrin-1).
(TIF)

**S8 Fig. Comparison of Netrin-1 sequence homology with synthetic peptides and identification of the heparin-binding region.** The specific residues involved in the heparin/heparan sulfate interaction are highlighted, and the mutations introduced to disrupt heparin binding are indicated.
(TIF)

**S9 Fig. Netrin-1 modulates preS1 internalization in response to EGF stimulation.** A: Serum-starved Dox-inducible Netrin-1-overexpressing HepG2-NTCP-YFP cells were incubated with or without 25 ng/mL Dox for 72 h, and then inoculated with the preS1-probe at 4°C for 1.5 h to allow surface binding. The cells were then incubated at 37°C for 6 h in the presence or absence of 100 ng/mL EGF to enable internalization. Ten fields of view from each sample were analyzed, and internalized preS1-probe signals were quantified and plotted. B: Representative fluorescence images of cells (preS1-probe, red; NTCP-YFP, green; nucleus, blue), shown at low (scale bar, 10 μm) and high (scale bar, 4 μm) magnification. Data shown represent the mean ± standard error of the mean from three independent experiments. Statistical testing was performed by two-way ANOVA with Tukey's multiple comparison test. ****$p < 0.0001$, ***$p < 0.001$, *$p < 0.05$, ns = not significant.
(TIF)

**S10 Fig. Netrin-1 impairs EGFR internalization.** A: Starved Dox-inducible Netrin-1-overexpressing HepG2-NTCP-YFP cells were incubated with or without 25 ng/mL Dox for 72 h, followed by treatment with 100 ng/mL EGF at 4°C for 30 min to allow EGF binding to the cell surface. After replacing the EGF-containing medium with starvation medium with or without 25 ng/mL Dox, the cells were shifted to 37°C to initiate EGFR internalization. The internalized EGFR signals were detected at 0, 5, 15, and 30 min, as well as at 1, 2, 4, and 6 h. The cells were then fixed and stained with an anti-EGFR antibody (red) and DAPI (blue), and imaged using confocal microscopy. Internalized EGFR signals were quantified using ImageJ from 10 randomly selected fields per sample and plotted in the graph. Cells not treated with EGF were used as a negative control. B: Representative confocal images at 0 and 30 min (EGFR, red; nucleus, blue) are shown at low (scale bar, 10 μm; left) and high (scale bar, 4 μm; right) magnification.
(TIF)

**S11 Fig. Netrin-1 binds to the ECD of EGFR via multiple domains.** Overexpression plasmids encoding EGFR-ECD-Flag and either wild-type or truncated forms of Netrin-1-HA were co-transfected into HepG2 cells. At 48 h post-transfection, the cells were harvested for immunoprecipitation with α-Flag antibody (for EGFR-ECD) or α-HA antibody (for Netrin-1) and detected by α-Flag antibody (for EGFR-ECD) or α-HA antibody (for Netrin-1).
(TIF)

**S12 Fig. Serum Netrin-1 levels in mice following administration of rNetrin-1 or PBS.** A: Serum Netrin-1 levels of blood samples collected at the timepoints indicated in Fig 7A were measured by ELISA specific for human Netrin-1. Week 0 samples were obtained before pump implantation, and week 1–5 samples were collected 1–5 weeks after HBV inoculation. Samples for weeks 3–5 were collected 48 h after the supplemental intravenous injections. B: A single intravenous bolus of rNetrin-1 (5 μg in 100 μL PBS) was administered to three mice via the tail vein. Blood samples were collected before administration and at 5 min, 30 min, 2 h, 4 h, 8 h, 24 h, and 48 h post administration. Serum Netrin-1 levels were measured by ELISA. Data shown represent the mean ± standard error of the mean from three independent experiments.

Statistical testing was performed by two-way ANOVA with Tukey's multiple comparison test. ****$p < 0.0001$, ***$p < 0.001$, *$p < 0.05$, ns = not significant.
(TIF)

**S1 Table. Oligo sequences used for cloning of shRNAs and qRT-PCR.**
(XLSX)

**S2 Table. Data that underlies this paper.**
(XLSX)

## Author contributions

**Conceptualization:** Masao Honda.

**Data curation:** Kazuhisa Murai.

**Formal analysis:** Ying Wang, Kazuhisa Murai.

**Funding acquisition:** Masao Honda.

**Investigation:** Ying Wang, Kazuhisa Murai, Atsuya Ishida, Narumi Kawasaki, Ying-Yi Li, Yuga Sato, Yutaro Miura, Kureha Takara, Lianghao Kong.

**Methodology:** Kazuyuki Kuroki, Ying-Yi Li.

**Project administration:** Masao Honda.

**Resources:** Kazuyuki Kuroki, Ying-Yi Li, Tetsuro Shimakami, Kouki Nio, Yuichiro Higuchi, Hiroshi Suemizu.

**Software:** Ying-Yi Li.

**Supervision:** Tetsuro Shimakami, Satoru Ito, Hiroshi Yanagawa, Shuichi Kaneko, Taro Yamashita, Masao Honda.

**Validation:** Atsuya Ishida, Narumi Kawasaki, Ying-Yi Li, Yuga Sato, Yutaro Miura, Kureha Takara, Lianghao Kong.

**Visualization:** Kazuhisa Murai, Masao Honda.

**Writing – original draft:** Ying Wang.

**Writing – review & editing:** Masao Honda.

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
