## [Decision Letter · Decision Letter 0]

5 Jun 2025

Netrin-1 inhibits the attachment and internalization of hepatitis B virus for hepatocyte infection

PLOS Pathogens

Dear Dr. Honda,

Thank you for submitting your manuscript to PLOS Pathogens. After careful consideration, we feel that it has merit but does not fully meet PLOS Pathogens's publication criteria as it currently stands. Therefore, we invite you to submit a revised version of the manuscript that explicitly addresses the points raised during the review process.

Please submit your revised manuscript within 60 days Aug 04 2025 11:59PM. If you will need more time than this to complete your revisions, please reply to this message or contact the journal office at plospathogens@plos.org. Please include the following items when submitting your revised manuscript:

We look forward to receiving your revised manuscript.

Kind regards,

Haitao Guo

Academic Editor

PLOS Pathogens

Robert Kalejta

Section Editor

PLOS Pathogens

Editor-in-Chief

PLOS Pathogens

PLOS Pathogens

orcid.org/0000-0002-7699-2064

**Journal Requirements:**

At this stage, the following Authors/Authors require contributions: Ying Wang, Kazuhisa Murai, Astuya Ishida, Narumi Kawasaki, Kazuyuki Kuroki, Ying Li, Yuga Sato, Yutaro Miura, Kureha Takara, Lianghao Kong, Tetsuro Shimakami, Kouki Nio, Yuichiro Higuchi, Hiroshi Suemizu, Satoru Ito, Hiroshi Yanagawa, Shuichi Kaneko, Taro Yamashita, and Masao Honda. Please ensure that the full contributions of each author are acknowledged in the "Add/Edit/Remove Authors" section of our submission form.

- TM on page: 18.

4) Your manuscript is missing the following sections: Abstract.  Please ensure all required sections are present and in the correct order. Make sure section heading levels are clearly indicated in the manuscript text, and limit sub-sections to 3 heading levels. An outline of the required sections can be consulted in our submission guidelines here:

https://journals.plos.org/plospathogens/s/submission-guidelines#loc-parts-of-a-submission 

5) Please upload all main figures as separate Figure files in .tif or .eps format. For more information about how to convert and format your figure files please see our guidelines: 

6) We notice that your supplementary Figures are included in the manuscript file. Please remove them and upload them with the file type 'Supporting Information'. Please ensure that each Supporting Information file has a legend listed in the manuscript after the references list.

7) Some material included in your submission may be copyrighted. According to PLOSu2019s copyright policy, authors who use figures or other material (e.g., graphics, clipart, maps) from another author or copyright holder must demonstrate or obtain permission to publish this material under the Creative Commons Attribution 4.0 International (CC BY 4.0) License used by PLOS journals. Please closely review the details of PLOSu2019s copyright requirements here: PLOS Licenses and Copyright. If you need to request permissions from a copyright holder, you may use PLOS's Copyright Content Permission form.

Potential Copyright Issues:

i) Figures 2A, 4A, 4J, and 5H. Please confirm whether you drew the images / clip-art within the figure panels by hand. If you did not draw the images, please provide (a) a link to the source of the images or icons and their license / terms of use; or (b) written permission from the copyright holder to publish the images or icons under our CC BY 4.0 license. Alternatively, you may replace the images with open source alternatives. See these open source resources you may use to replace images / clip-art:

8) We note that your Data Availability Statement is currently as follows: "All data are fully available, without restriction.". Please confirm at this time whether or not your submission contains all raw data required to replicate the results of your study. Authors must share the “minimal data set” for their submission. PLOS defines the minimal data set to consist of the data required to replicate all study findings reported in the article, as well as related metadata and methods (https://journals.plos.org/plosone/s/data-availability#loc-minimal-data-set-definition).

9) Please provide a detailed Financial Disclosure statement. This is published with the article. It must therefore be completed in full sentences and contain the exact wording you wish to be published.

1) Please clarify all sources of financial support for your study. List the grants, grant numbers, and organizations that funded your study, including funding received from your institution. Please note that suppliers of material support, including research materials, should be recognized in the Acknowledgements section rather than in the Financial Disclosure

2) State the initials, alongside each funding source, of each author to receive each grant. For example: "This work was supported by the National Institutes of Health (####### to AM; ###### to CJ) and the National Science Foundation (###### to AM)."

3) State what role the funders took in the study. If the funders had no role in your study, please state: "The funders had no role in study design, data collection and analysis, decision to publish, or preparation of the manuscript."

4) If any authors received a salary from any of your funders, please state which authors and which funders..

10) Your current Financial Disclosure states, "The author(s) received no specific funding for this work.".

However, your funding information on the submission form indicates Funder Name: Japan Agency for Medical Research and Development; JP25fk0310539, JP23fk0310514, JP23fk0210129 and JP23fk0210140 to Dr Masao Honda. 

Please indicate by return email the full and correct funding information for your study and confirm the order in which funding contributions should appear. Please be sure to indicate whether the funders played any role in the study design, data collection and analysis, decision to publish, or preparation of the manuscript.

11) Please send a completed 'Competing Interests' statement, including any COIs declared by your co-authors. If you have no competing interests to declare, please state "The authors have declared that no competing interests exist". Otherwise please declare all competing interests beginning with the statement "I have read the journal's policy and the authors of this manuscript have the following competing interests:"

**Reviewers' Comments:**

Reviewer's Responses to Questions

**Part I - Summary**

Reviewer #1: The authors' group previously reported that endothelial lipase (LIPG) functions as a host factor that facilitates the attachment of HBV virions to the hepatocyte membrane and their subsequent internalization. They also identified Netrin-1 as a potential LIPG-interacting protein and demonstrated that synthetic peptides based on the Netrin-1 sequence can inhibit HBV infection. In the current study, they utilized a Dox-1 dose-dependent Netrin-1 induction cell model, primary PXB cells, and humanized hepatocyte chimeric (TK-NOG) mice to conduct a more in-depth investigation into the molecular mechanisms underlying Netrin-1's inhibitory effects on HBV infection. Overall, the present study is potentially interesting. However, from my perspective, there are several confusing aspects in the logic of the manuscript, which are detailed in the major concerns section below.

Reviewer #2: In this manuscript, the authors have conducted a series of extensive experiments to support their central hypothesis. Although the finding is not novel as the LIPG-Nterin-1, and LIPG-LHBs interactions have been reported previously, however, this follow up study is indeed significant as it focused on detailed mechanistic insights. Notably, authors appear to have undertaken a thorough effort in generating multiple expression constructs for various mutants. Each mutant seems to be thoughtfully designed to test specific aspects of their hypothesis, and the corresponding results are generally in alignment with those aims. However, the study fell short in convincingly addressing three key aspects described below.

Reviewer #3: Ying Wang and colleagues report in this manuscript that Netrin-1, a secreted laminin-related protein, interacts with endothelial lipase (LIPG) to inhibit HBV attachment to hepatocytes in a heparan sulfate proteoglycans (HSPGs)-dependent manner. In addition, Netrin-1 can also bind to the extracellular domain of the EGFR to abrogate NTCP-EGFR complex formation and internalization of HBV into hepatocytes. The authors also demonstrated that a hepatocyte-targeting Netrin-derived LIPG binding peptides, 10M-LNTSN and its derivative 10ML-B28, efficiently inhibited HBV infection in primary human hepatocytes in culture and in vivo in PXB mice.

Overall, the study reveals key molecular features of HBV infection of hepatocytes. The experiments are well conceived and executed. The conclusions are supported by the data presented. My concerns are generally minor.

**Part II – Major Issues: Key Experiments Required for Acceptance**

Reviewer #1: 1. The authors propose that the extracellular secreted protein LIPG facilitates the attachment of HBV to the hepatocyte surface and promotes viral internalization. On the other hand, hepatocytes (PXB cells, Fig. 1B) also express the LIPG-binding protein Netrin-1, which inhibits the aforementioned process. Netrin-1 may also inhibit viral endocytosis by suppressing the interaction between NTCP and EGFR. Given this, how are these processes regulated temporally and spatially during natural HBV infection? Does the expression of Netrin-1 represent a host cell's antiviral defense mechanism following viral infection?

Based on the presented data, neither Huh7 nor HepG2-derived tumor cell lines express endogenous Netrin-1. Even in HBV-expressing HepG2.2.15 cells, Netrin-1 expression cannot be detected (Fig. 1D). Does this contradict the mRNA quantification data shown in Fig. 1B? Does it imply that even very low levels of Netrin-1 expression can strongly inhibit HBV infection? Given that Netrin-1 expression is barely detectable at the protein level, is the proposed mechanism of competitive protein binding involving Netrin-1 reasonable? The authors should also determine the endogenous expression levels of Netrin-1 in PXB cells.

2. Fig. 2F: LIPG appears to be a highly sticky protein, interacting with various functional domains of Netrin-1. Is there a common structural basis for these interactions? Particularly noteworthy is that LIPG also binds to the L protein of HBV (Fig. 4B, C). Could additional experiments be designed, beyond co-immunoprecipitation (Co-IP) assays in cells, to further demonstrate direct interactions between purified proteins? Additionally, are all the proteins shown in Fig. 2G secreted outside the cell, thereby enabling their role in inhibiting viral attachment?

3. In my view, the experimental design in Fig. 7 is highly problematic, and is the aspect of this study that concerns me the most. The preceding logic is that Netrin-1 inhibits HBV infection of hepatocytes through various competitive protein-protein interactions on the cell surface. However, the construct designed in Fig. 7A surprisingly introduces an endosomal cleavage site, a fusogenic peptide, and an NLS peptide. What are their functions? Are they still intended to inhibit viral entry at the cell surface? Additionally, using PBS as a control in the in vivo experiments is clearly unreasonable. The authors should consider whether ASGR monoclonal antibodies also affect HBV infection of hepatocytes. Moreover, the naming of the LIPG peptide is also very confusing.

Reviewer #2: 1. Concerns about relevance to natural infection: A more fundamental concern lies in the conceptual framing of the hypothesis. Even if we assume that the authors' experimental observations are accurate, it is difficult to reconcile the proposed mechanism with the known biology of viral infection. The central hypothesis posits that Netrin-1 inhibits HBV internalization and attachment, which are events that occur extracellularly or at the cell surface. However, most of the experiments described in the manuscript focus on intracellular interactions—specifically, protein-protein interactions, displacement assays, and competition experiments within the cell. More importantly, Netrin-1 is a secretory protein, as the authors themselves acknowledge in the manuscript. In light of this, the authors need to provide a strong justification for why the intracellular interactions among LIPG, LHBs, and Netrin-1 are expected to yield biologically relevant insights.

This raises an important question: how do these intracellular events mechanistically relate to the inhibition of viral entry, which is proposed to occur at the cell surface? Before delving further into the technical details of the experiments, the authors need to more clearly explain how their hypothesis and current data connect to the natural course of viral infection. Without this clarification, the physiological relevance of the findings remains uncertain.

2. The proposed model should be defined more clearly: The authors should clearly distinguish whether the effects of ectopic expression of Netrin-1 (and its various variants) are exclusively cis-acting or if they can also function in trans. Given that Netrin-1 is a secreted protein and the overall hypothesis appears to operate in the extracellular environment (as depicted in Figures 2A, 4J, and 5H), it is important to experimentally address this distinction. One way to test this would be to use cells overexpressing Netrin-1 and assess a relevant effect, such as viral attachment or internalization (that authors have done it already). These Netrin-1 overexpressing cells could be co-cultured with control cells (which do not overexpress Netrin-1) in a transwell system. If the authors’ hypothesis is correct and Netrin-1 acts in trans, the control cells should partially adopt the phenotype observed in Netrin-1 overexpressing cells due to secreted signaling molecules. Conversely, if the effect is strictly cis-acting, the control cells would remain unaffected (for example no effect on viral internalization), and the phenotype would only be observed in the cells directly overexpressing Netrin-1. This experiment would provide critical insight into the mechanism of Netrin-1 action and help clarify whether its signaling is cell-autonomous or involves intercellular communication.

3. Lack of details and experimental clarity: One of the weaker aspects of the manuscript is the figure legends, which lack sufficient details and consistency. Furthermore, there are cases where the claims made in the text are not convincingly supported by the data presented in the figures. There are also multiple instances where the authors could have used simpler, more straightforward experimental strategies to validate their conclusions. Instead, they have chosen more complex approaches, which may not enhance the scientific value of the study and, in some cases, obscure the interpretation of the data.

Reviewer #3: No.

**Part III – Minor Issues: Editorial and Data Presentation Modifications**

Reviewer #1: 1. Fig. 1D: cccDNA serves as the template for viral replication and typically exists in low copy numbers within cells. From a stoichiometric perspective, the intensity of the cccDNA band should not appear similar to that of the rcDNA (relaxed circular DNA) band. Additionally, why were the bands for the replication intermediates, such as Dsl DNA (double-stranded linear DNA) and SS DNA (single-stranded DNA), not observed in the figure?

2. Lines 162–167: The description of HSPG-dependent and HSPG-independent HBV attachment in this section is confusing. The results in Fig. 2C actually suggest that the anti-HBV mechanism of Netrin-1 is downstream of Heparanase activity, meaning it is HSPG-dependent.

3. Fig. 5D: The image in Fig. 5D lacks statistical analysis. Additionally, why were the NTCP signal and preS1 probe signal not merged for analysis?

4. Fig. 6D: EGF stimulation induces EGFR internalization, which is known to be partially degraded via the lysosomal pathway. However, this well-documented phenomenon does not appear to be reflected in the results shown in Fig. 6. Additionally, it is suggested to directly analyze the inhibitory effect of Netrin-1 on EGF-stimulated EGFR internalization using image data.

Reviewer #2: 1. Why different scheme was adopted for Fig 1A and S1 Fig A (detection times were day 10 and day 17 respectively). Did authors check the effect of exogenous addition of rNetrin-1 (what was done in fig 1E) on HBV infection using PXB cells (like fig 1 C and D). If the central hypothesis is true, the authors should see some effect which will be opposite to what is reported in fig 1C and 1D.

2. For the figure 1 A and B, did authors check the protein depletion using western blot? What is the authors’ observation on endogenous expression of Netrin-1? Can authors provided western blot for cells they used to see how abundant this protein is? It will be very informative to know how much Netrin-1 is produced, retained within the cells, and secreted out.

3. Which antibody was used in fig. 1F and 1j? Current labeling says “Netrin-1” and respective legend doesn’t provide any detials. If the Netrin-1 construct is tagged, such information shouldbe provided in the figure, if it was anti-Netrin-1 antibody then discuss why no band is visible in the control lane.

4. Control lane of fig 1M, in which authors have used anti-Netrin-1 antibody, shows that endogenous level of this protein is very low as hardly any band is detectable. This further substantiates the need to address the second half of the above comment #2.

5. Line 196 and 198, the figure 1E should be figure 2E.

6. Figure 4I needs normalization control for membrane protein. Authors can choose any known protein that resides in membrane fraction and show it as a control to prove that membrane fractions were loaded equally.

7. Figure 5 needs some clarification. I am not sure if the experiment 5A is fully relevant to the hypothesis authors proposed here. The best answers will be provided by experiment when Netrin1 is exogenously added to these cells (and not ectopically expressed) and then see if Netrin affects 1068 p-EGFR level. If the authors believe the model proposed in fig. 5H, then exogenous addition of purified Netrin-1 should also affect EGFP 1068 phosphorylation or EGFR-NTCP interaction as shown in 5G. Fortunately, the authors seem to have access to purified rNetrin-1 protein that they have used in fig. 1 E and elsewhere.

8. Figure 6, it will further strengthen the hypothesis if authors can perform few of these experiments (wherever feasible) using exogenously added r-Netrin-1.

9. More details should be provided for the in vivo treatment exp. How many micer in each group? What did the viremia look like prior to treatment during the 7 weeks? These are immunodeficient mice and should not mount any antiviral response so it is unclear about the ALT differences between the two groups. Why did the ALT levels went up over during treatment with PBS? Given the peptide treatment mainly targeted viral entry, one would expect the viremia to rebound to the control level in the treated mice after treatment. The authors should have monitored these mice after the treatment.

10. The Discussion section requires substantial improvement. In its current form, it largely restates the results without providing deeper analysis or interpretation. The authors should incorporate additional relevant information to enhance this section. For example, the level of Netrin-1 mRNA in HBV-positive cohorts has been reported previously (see Plissonnier et al., PLoS Biology, 2016). Furthermore, more attention should be given to the nature of the current hypothesis. It would be valuable to clarify whether the proposed Netrin-1 signaling functions primarily in a cis-acting or trans-acting manner. Finally, the authors should acknowledge the limitations of their current model. What questions remain unanswered? What aspects should be addressed in future studies to advance understanding of this signaling pathway? Adding these elements will greatly strengthen the Discussion and provide clearer context for the study’s implications.

Reviewer #3: 1. For the experimental results presented in Fig. 1A to 1D, while shRNA#1 less efficiently reduced Netrin-1 mRNA in PXB cells (Fig. 1B), it more efficiently increased intracellular HBV DNA than shRNA#2. Why?

2. Fig. S1. Lack of data to demonstrate the shRNA knockdown efficiency on their respective target RNA.

3. For the experimental results presented in Fig. 1G, the concentration of recombinant Netrin-1 should be indicated in the Figure or figure legend.

4. For the experiment presented in Fig. 2C, how long the cells were treated with Dox to induce Netrin-1 expression? This procedure should be included in the experimental schedule diagram on the left panel.

5. For the data presented in Fig. 2G and H, have you determined the subcellular localization of the three Netrin-1 fragments? Whether the failure of Netrin-1 VI fragment to inhibit HBV attachment is due to its inability to be transported onto the cell surface?

PLOS authors have the option to publish the peer review history of their article (what does this mean? ). If published, this will include your full peer review and any attached files.

**Do you want your identity to be public for this peer review?** For information about this choice, including consent withdrawal, please see our Privacy Policy .

Reviewer #1: No

Reviewer #2: No

Reviewer #3: No

**Figure resubmission:**

**Reproducibility:**



---

## [Decision Letter · Decision Letter 1]

17 Sep 2025

Netrin-1 inhibits the attachment and internalization of hepatitis B virus for hepatocyte infection

PLOS Pathogens

Dear Dr. Honda,

Please submit your revised manuscript within 60 days Nov 16 2025 11:59PM. If you will need more time than this to complete your revisions, please reply to this message or contact the journal office at plospathogens@plos.org. Please include the following items when submitting your revised manuscript:

We look forward to receiving your revised manuscript.

Kind regards,

Haitao Guo

Academic Editor

PLOS Pathogens

Robert Kalejta

Section Editor

Editor-in-Chief

PLOS Pathogens

orcid.org/0000-0003-2946-9497

Editor-in-Chief

PLOS Pathogens

orcid.org/0000-0002-7699-2064

**Journal Requirements:**

At this stage, the following Authors/Authors require contributions: Ying Wang, Kazuhisa Murai, Astuya Ishida, Narumi Kawasaki, Kazuyuki Kuroki, Ying-Yi Li, Yuga Sato, Yutaro Miura, Kureha Takara, Lianghao Kong, Tetsuro Shimakami, Kouki Nio, Yuichiro Higuchi, Hiroshi Suemizu, Satoru Ito, Hiroshi Yanagawa, Shuichi Kaneko, Taro Yamashita, and Masao Honda. Please ensure that the full contributions of each author are acknowledged in the "Add/Edit/Remove Authors" section of our submission form.

2) Please confirm whether your study includes live participants, if so please insert an Ethics Statement at the beginning of your Methods section, under a subheading 'Ethics Statement'. It must include:

i) The full name(s) of the Institutional Review Board(s) or Ethics Committee(s)

ii) The approval number(s), or a statement that approval was granted by the named board(s).

3) Your current Financial Disclosure states, "The author(s) received no specific funding for this work."

However, your funding information on the submission form indicates receiving funds. Please ensure that the funders and grant numbers match between the Financial Disclosure field and the Funding Information tab in your submission form. Note that the funders must be provided in the same order in both places as well.

Please amend your detailed Financial Disclosure statement. This is published with the article. It must therefore be completed in full sentences and contain the exact wording you wish to be published.

1) Please clarify all sources of financial support for your study. List the grants, grant numbers, and organizations that funded your study, including funding received from your institution. Please note that suppliers of material support, including research materials, should be recognized in the Acknowledgements section rather than in the Financial Disclosure

2) State the initials, alongside each funding source, of each author to receive each grant. For example: "This work was supported by the National Institutes of Health (####### to AM; ###### to CJ) and the National Science Foundation (###### to AM)."

3) State what role the funders took in the study. If the funders had no role in your study, please state: "The funders had no role in study design, data collection and analysis, decision to publish, or preparation of the manuscript."

4) If any authors received a salary from any of your funders, please state which authors and which funders.

4) If your research concerns only data provided within your submission, please update your Data Availability Statement in the online submission form to "All relevant data are within the paper and its Supporting information files" as you stated in your manuscript.

**Reviewers' Comments:**

Reviewer's Responses to Questions

**Part I - Summary**

Reviewer #1: No more comments.

Reviewer #2: In this revision, the authors made a reasonable effort to respond to the previous reviews. The authors performed additional exp to demonstrate that the secreted Netrin indeed blocked HBV infection in a trans well co-culture model, supporting its effect on HBV binding/entry to the cells. However, it is not clear why the authors continued to push for additional intracellular effects of Netrin on HBV infection, which are not well conceived nor supported by existing data. It is also not clear why the authors designed such a complicated scheme for Netrin-related peptide production. It would have been much easier just to produce recombinant Netrin without the additional cargos to assess its anti-HBV activities. The additional anti-ASGR cargo just complicated the interpretation of the data. A better negative control would be just the anti-ASGP. The authors should have explored the in vivo study further, the data of which were not very convincing. The authors should also temper their conclusion based on their data. Otherwise, the experiments and data are rather extensive and thorough.

Reviewer #3: The authors have adequately addressed my comments on the previous version of this manuscript!

**Part II – Major Issues: Key Experiments Required for Acceptance**

Reviewer #1: The authors have addressed most of the questions I raised in the revised manuscript and made the necessary revisions. However, I still have significant concerns about the design of the in vivo experiment in the final section. The authors' peptide has multiple antiviral objectives, yet only PBS was used as a control. This makes it impossible to discern whether the primary mechanism by which the peptide exerts its antiviral effects is through the cell surface entry mechanism discussed earlier or through some post-entry step. Furthermore, ASGR undergoes efficient endocytosis under ligand stimulation. If this is the case, the antiviral effects of the peptide would not logically align with the previously discussed mechanism of inhibiting viral entry.

Reviewer #2: 1. Simplify and streamline the MS to focus on the likely mechanism of action of Netrin's anti-HBV effect.

2. Expand experimentally the part on the recombinant Netrin and its effect in vitro and in vivo using a proper control. Otherwise remove that section.

Reviewer #3: No.

**Part III – Minor Issues: Editorial and Data Presentation Modifications**

Reviewer #1: No more comments.

Reviewer #2: (No Response)

Reviewer #3: No.

PLOS authors have the option to publish the peer review history of their article (what does this mean? ). If published, this will include your full peer review and any attached files.

**Do you want your identity to be public for this peer review?** For information about this choice, including consent withdrawal, please see our Privacy Policy .

Reviewer #1: No

Reviewer #2: No

Reviewer #3: No

**Figure resubmission:**

**Reproducibility:**



---

## [Decision Letter · Decision Letter 2]

1 Dec 2025

Dear Dr Honda,

We are pleased to inform you that your manuscript 'Netrin-1 inhibits the attachment and internalization of hepatitis B virus for hepatocyte infection' has been provisionally accepted for publication in PLOS Pathogens.

Best regards,

Haitao Guo

Academic Editor

PLOS Pathogens

Robert Kalejta

Section Editor

PLOS Pathogens

Sumita Bhaduri-McIntosh

Editor-in-Chief

PLOS Pathogens

orcid.org/0000-0003-2946-9497

Michael Malim

Editor-in-Chief

PLOS Pathogens

orcid.org/0000-0002-7699-2064

Reviewer Comments (if any, and for reference):

Reviewer's Responses to Questions

**Part I - Summary**

Reviewer #1: The authors have adequately addressed the questions I raised, and I have no further concerns regarding the revised manuscript.

Reviewer #2: In the revision, the authors had simplified the in vivo exp, which is more consistent with the proposed anti-HBV mechanism of Netrin-1. This reviewer does not have any further comments.

**Part II – Major Issues: Key Experiments Required for Acceptance**

Reviewer #1: No more comments.

Reviewer #2: (No Response)

**Part III – Minor Issues: Editorial and Data Presentation Modifications**

Reviewer #1: No more. comments

Reviewer #2: (No Response)

PLOS authors have the option to publish the peer review history of their article (what does this mean? ). If published, this will include your full peer review and any attached files.

**Do you want your identity to be public for this peer review?** For information about this choice, including consent withdrawal, please see our Privacy Policy .

Reviewer #1: No

Reviewer #2: No

---

## [Editor Report · Acceptance letter]

Dear Dr Honda,

We are delighted to inform you that your manuscript, " 

Netrin-1 inhibits the attachment and internalization of hepatitis B virus for hepatocyte infection," has been formally accepted for publication in PLOS Pathogens.

Best regards,

Sumita Bhaduri-McIntosh

Editor-in-Chief

PLOS Pathogens

orcid.org/0000-0003-2946-9497

Michael Malim

Editor-in-Chief

PLOS Pathogens

orcid.org/0000-0002-7699-2064